# Improved Convergence of Score-Based Diffusion Models via Prediction-Correction

**Francesco Pedrotti**  *francesco.pedrotti@ista.ac.at*
*Institute of Science and Technology Austria*

**Jan Maas**  *jan.maas@ista.ac.at*
*Institute of Science and Technology Austria*

**Marco Mondelli**  *marco.mondelli@ista.ac.at*
*Institute of Science and Technology Austria*

**Reviewed on OpenReview:** *https://openreview.net/forum?id=0zKvH7YiAq*

## Abstract

Score-based generative models (SGMs) are powerful tools to sample from complex data distributions. Their underlying idea is to *(i)* run a forward process for time $T_1$ by adding noise to the data, *(ii)* estimate its score function, and *(iii)* use such estimate to run a reverse process. As the reverse process is initialized with the stationary distribution of the forward one, the existing analysis paradigm requires $T_1 \to \infty$. This is however problematic: from a theoretical viewpoint, for a given precision of the score approximation, the convergence guarantee fails as $T_1$ diverges; from a practical viewpoint, a large $T_1$ increases computational costs and leads to error propagation. This paper addresses the issue by considering a version of the popular *predictor-corrector* scheme: after running the forward process, we first estimate the final distribution via an inexact Langevin dynamics and then revert the process. Our key technical contribution is to provide convergence guarantees which require to run the forward process *only for a fixed finite time* $T_1$. Our bounds exhibit a mild logarithmic dependence on the input dimension and the subgaussian norm of the target distribution, have minimal assumptions on the data, and require only to control the $L^2$ loss on the score approximation, which is the quantity minimized in practice.

## 1 Introduction

Score matching models (Song & Ermon, 2019; 2020) and diffusion probabilistic models (Sohl-Dickstein et al., 2015; Ho et al., 2020) – recently unified into the single framework of score-based generative models (SGMs) (Song et al., 2020) – have shown remarkable performance in sampling from unknown complex data distributions, achieving the state of the art in image (Song et al., 2020; Dhariwal & Nichol, 2021) and audio (Popov et al., 2021; Kong et al., 2021; Chen et al., 2021b) generation; see also the recent surveys Yang et al. (2023); Croitoru et al. (2023). The idea is to gradually perturb the data by adding noise, and then to learn to revert the process. Both the forward process that adds noise and the reverse process can be described by a stochastic differential equation and, specifically, the reverse process is defined in terms of the *score function* (*i.e.*, the gradient of the logarithm of the perturbed density at all noise scales, see Section 3 for details). This time-dependent score function can be learned with a neural network, using efficient techniques such as sliced score matching (Song et al., 2020) or denoising score matching (Vincent, 2011). Then, to start the reverse process, one would ideally need to sample from the perturbed distribution, which is in principle unknown: instead, one runs the forward process for a long enough time $T_1$ so that the perturbed distribution $p_{T_1}$ is well approximated by the stationary distribution $\pi$, which is known and can be readily sampled from.

A central theoretical question is to understand the quality of the sampling, *i.e.*, measure a distance between the output distribution of the reverse process and the true one. Three sources of error are given by *(i)*

starting the reverse process from the stationary distribution $\pi$, rather than from the perturbed distribution $p_{T_1}$, *(ii)* approximating the score function (*e.g.*, with a neural network), and *(iii)* discretizing the reverse stochastic differential equation. The quantitative characterization of such errors has been carried out in a number of recent papers, see Song et al. (2021a); Kwon et al. (2022); Lee et al. (2022); Chen et al. (2023b; 2022a); Bortoli (2022) and Section 2. However, to achieve convergence of the output distribution to the ground truth, this line of work requires to run the forward process for $T_1 \to \infty$. This is due to the first source of error mentioned above, *i.e.*, the approximation of $p_{T_1}$ with $\pi$. At the same time, large values of $T_1$ amplify the other two sources of error and are also responsible for an increased computational cost in the training procedure, because of the need to approximate the score function on a large time interval $[0, T_1]$. Thus, there appears to be a subtle trade-off between the precision in the score approximation and the running time $T_1$: on the one hand, one needs to take $T_1 \to \infty$ so that $p_{T_1}$ approaches $\pi$; on the other hand, for a given precision in the score, the convergence guarantees fail as $T_1 \to \infty$,[1] which highlights the instability of existing results. For these reasons, it is of great interest to characterize an appropriate time $T_1$ where to stop the forward process (Yang et al., 2023, Sec. 8).

**Main contribution.** In this work, we address the trade-off in the choice of the running time $T_1$ of the forward process by considering a variant of the predictor-corrector methods of Song et al. (2021b). More precisely, after obtaining $p_{T_1}$ via the forward process, we sample from $p_{T_1}$ via an inexact Langevin dynamics that leverages the approximation of the score at time $T_1$. Then, we use the resulting sample to start a standard reverse process. Our main convergence result (Theorem 4.1) focuses on the deterministic reverse process, which has the form of an ordinary differential equation (ODE), and analyzes the proposed algorithm: in particular, it provides convergence guarantees in Wasserstein distance for a vast class of data distributions and under realistic assumptions on the score estimation, which are compatible with the training loss used in practice. We highlight that our bounds require a perturbation time $T_1$ that only depends logarithmically on the dimension of the space and on the subgaussian norm of the target distribution, and *not* on the desired sampling precision. The mild logarithmic dependence suffices to ensure the regularity – in the form of a log-Sobolev inequality – of the perturbed measure $p_{T_1}$, which in turn allows the inexact Langevin dynamics to converge exponentially fast.

Our analysis improves upon earlier bounds in Wasserstein distance (Kwon et al., 2022) by removing both the need for $T_1 \to \infty$ and an assumption on the score estimator (more precisely, on its one-sided Lipschitz constant, see the discussion after Theorem 4.1 for details). This comes at the cost of requiring a control on a loss function for the score estimate at time $T_1$, which is stronger than the usual $L^2$ loss. To address this issue, we exploit the fast convergence of the forward process to its stationary distribution in order to correct the score estimator, which in turn allows us to translate upper bounds for the $L^2$ loss into upper bounds for the stronger loss (Theorem 4.2). Finally, when considering instead a stochastic reverse process (SDE), we show how the algorithm is compatible with the existing analyses for the convergence of the discretized reverse SDE in information-divergence metrics. This allows us to deduce convergence guarantees in total variation distance for a discretization of the algorithm. Consequently, we highlight some advantages over previous results that arise from the choice of a fixed perturbation time $T_1$, related again to the decreased computational cost in the training procedure and to the stability of the error bounds.

**Paper organization.** Section 2 discusses related works. Section 3 sets up the technical framework by recalling the formal description of SGMs. Section 4 presents our main contribution: after describing the algorithm, we state the convergence result in Wasserstein distance. Section 4.1 contains a sketch of the proofs, with the full arguments deferred to the appendix. Our main convergence results in Section 4 are stated in continuous time and in Wasserstein distance. Then, Section 5 contains a discussion about discretizations of the algorithm, as well as a convergence result in total variation distance. Section 6 concludes the paper with some final remarks.

---

[1]See, *e.g.*, Chen et al. (2023b, Thm. 2) and note that the third term in the bound diverges when $\varepsilon_{\text{score}}$ is fixed and $T \to \infty$.

## 2 Related work

The empirical success of SGMs has led to extensive research aimed at providing theoretical guarantees on their performance. Specifically, the goal is to give upper bounds on the distance between the true data distribution $p$ and the output distribution $p_\theta$ of the sampling method. Adopting the description of forward and reverse process in terms of stochastic differential equations (Song et al., 2021b), an upper bound on the KL-divergence $D_{\mathrm{KL}}(p \,\|\, p_\theta)$ is provided in Song et al. (2021a): under some regularity assumptions, this KL-divergence goes to 0, as $T_1 \to \infty$ and the score approximation error vanishes. In a similar vein, using the theory of optimal transport instead of stochastic tools, an upper bound on the Wasserstein distance[2] $W_2(p, p_\theta)$ is provided in Kwon et al. (2022). In many important situations, results in Wasserstein distance are more meaningful than in KL-divergence or total variation: for example, under the manifold hypothesis, it is not possible to obtain non-trivial convergence guarantees in those metrics, as one has $D_{\mathrm{KL}}(p \,\|\, p_\theta) = \infty$ and $\mathrm{TV}(p, p_\theta) = 1$ (cf. Bortoli (2022); Chen et al. (2023b)). However, to obtain convergence, Kwon et al. (2022) imposes a strong assumption on the score estimator. A line of work has focused on the discretization of the reverse stochastic differential equation. Specifically, convergence in Wasserstein distance of order 1 is provided in Bortoli (2022) under the manifold hypothesis, but the results depend poorly on important parameters, such as the sampling precision, the input dimension and the diameter of the support of $p$. The works Lee et al. (2023); Chen et al. (2023b; 2022a) provide convergence guarantees for general distributions in KL-divergence and total variation (which is weaker by Pinsker's inequality). From these results, bounds in Wasserstein distance are also deduced for some classes of distributions, including bounded support ones. This improves upon Bortoli (2022), but the improvement requires an extra projection step and comes at the cost of a worse dependence on the problem's parameters in comparison with the bounds in KL-divergence and total variation. Other works have provided convergence results for SGMs, but they suffer from at least one of the following drawbacks (Chen et al., 2023b): *(i)* non-quantitative bounds (Pidstrigach, 2022; Bortoli et al., 2021) or poor dependence on important problem parameters (Block et al., 2022), *(ii)* strong assumptions on the data distribution, typically in the form of a functional inequality (Lee et al., 2022; Yingxi Yang & Wibisono, 2023), and *(iii)* strong assumptions on the score estimation error, such as a uniform pointwise control (Bortoli et al., 2021), which is not observed in practice (Zhang & Chen, 2023).

In most of these works, to guarantee convergence of $p_\theta$ to $p$, it is necessary to have $T_1 \to \infty$. In contrast, Bortoli et al. (2021) introduces a different approach based on solving the Schrödinger bridge problem (see also Chen et al. (2022b); Shi et al. (2022); Song (2022)), which allows for a finite time perturbation of the target measure; however, this strategy does not come with quantitative convergence results under realistic assumptions. The work Franzese et al. (2023) adopts an approach closer to ours: an auxiliary model is used to bridge the gap between the limiting distribution of the forward process and the true perturbed distribution, which is then followed by a standard reverse process. However, the design of the auxiliary model appears to be *ad hoc* for the data distribution, and a theoretical convergence result for general data distributions is missing. Our algorithm can also be considered as an instance of the predictor-corrector approach of Song et al. (2021b): there, the authors suggest to alternate one step of the reverse process with a few steps of a corrector method based on the score function, such as Langevin dynamics, and provide extensive empirical evidence showing an improved performance. In this work, we consider instead the case where all the corrector steps (Langevin dynamics) are performed at the beginning, and they are then followed by a standard (non-corrected) reverse process. We remark that error bounds for (a different variant of) the predictor-corrector schemes were first provided in Lee et al. (2022). The results therein, however, impose strong assumptions on the data distribution, in the form of a log-Sobolev inequality; the log-Sobolev constant enters crucially in the derived bounds, which depend polynomially on it, and not just logarithmically.

**Concurrent work.** The concurrent work Chen et al. (2023a) also studies the performance of a predictor corrector scheme. Instead of our two-stage algorithm, Chen et al. (2023a) considers an implementation closer to Song et al. (2021b), that alternates deterministic predictor steps with corrector steps based on Langevin dynamics. For this method, the authors provide convergence guarantees in total variation distance:

---

[2]For general data distributions $\mu, \nu$, there is no relation between $D_{\mathrm{KL}}(\nu \,\|\, \mu)$ and $W_2(\mu, \nu)$, therefore the results in KL-divergence cannot be translated trivially to results in $W_2$, and vice versa. In particular, the analysis in $W_2$ seems to be more challenging, because of an expansive term in the reverse process (cf. Chen et al. (2023b, Sec. 4)).

interestingly, they show that when the corrector steps are based on the *underdamped Langevin dynamics* (rather then the classical overdamped version), it is possible to obtain a better dependence on the dimension $d$. Contrary to our results, however, the error bounds in Chen et al. (2023a) still require $T_1 \to \infty$ to obtain convergence, with analogous disadvantages as we discussed for pre-existing results.

After the first version of our paper appeared online, further progress on the theoretical study of SGMs has been made. The work Li et al. (2023) obtains convergence guarantees for both the standard reverse SDE and the deterministic reverse ODE (without corrector), using an approach based on studying directly discrete time methods rather than controlling the errors in approximating the continuous time dynamics. The paper Benton et al. (2023) improves the dimension dependence of convergence guarantees for the reverse SDE, when one does not assume Lipschitzness of $\nabla \log p_t$, by exploiting a connection with stochastic localization (Eldan, 2013; Montanari, 2023). The work Conforti et al. (2023) proves convergence in KL-divergence without early stopping when replacing the classical Lipschitzness assumption on $\nabla \log p_t$ with finiteness of the relative Fisher information.

In general, these works can be considered complementary to the present contribution, in that their analysis techniques can be combined with our methods to provide convergence results for the two-stage algorithm under a set of different assumptions, gaining the advantage of a fixed perturbation time $T_1$. We refer the reader to Section 5 for an example of this.

## 3 Preliminaries

To sample from an unknown data distribution $p$ on $\mathbb{R}^d$, the framework of SGMs consists in perturbing $p$ through a forward process that converges to a known prior distribution and then approximately reverse this process. The forward process is described by a stochastic differential equation (SDE)

$$\begin{cases} X_0 \sim p, \\ dX_t = f(t, X_t)\, dt + g(t)\, dB_t, \end{cases} \tag{3.1}$$

where $B_t$ is a standard Brownian motion, $f$ and $g$ are sufficiently smooth, and the SDE is run until some time $T_1 > 0$. The law of $X_t$, denoted by $p_t$, correspondingly solves the Fokker–Planck equation

$$\begin{cases} p_0 = p, \\ \partial_t p_t + \nabla \cdot \left[ p_t \left( f(t, x) - \frac{g(t)^2}{2} \nabla \log p_t(x) \right) \right] = 0. \end{cases} \tag{3.2}$$

Remarkably, under some regularity conditions, this SDE admits a reverse process, in the sense that, for any smooth function $M \colon [0, T_1] \to \mathbb{R}_{\geq 1}$, the process $(U_t)_t$ defined by

$$\begin{cases} U_0 \sim p_{T_1}, \\ dU_t = -f(T_1 - t, U_t)\, dt + \frac{M(t)}{2} g(T_1 - t)^2 \nabla \log p_{T_1 - t}(U_t)\, dt + \sqrt{M(t) - 1}\, g(T_1 - t)\, dB_t, \end{cases} \tag{3.3}$$

is such that $U_{T_1} \sim p$. Usual choices are $M(t) \equiv 2$ or $M(t) \equiv 1$ and, considering the latter, the reverse process is deterministic, except for its initialization, see Song et al. (2021b, Sec. 4.3). Below, we will first focus on $M \equiv 1$ for simplicity, but similar results can be readily deduced for general $M(t)$.

To simulate the reverse SDE (3.3) and sample from $p$, one needs to *(i)* approximately sample from $p_{T_1}$ to initialize the backward process, and *(ii)* estimate the score function with $s_\theta(t, \cdot) \approx \nabla \log p_t$ for $t \in [0, T_1]$. For the first point, one chooses $T_1$ big enough so that $p_{T_1}$ is close to a known distribution $\pi$ and samples from $Y_0 \sim \pi$. For the second point, one learns a function $s_\theta(t, x)$ that approximates $\nabla \log p_t(x)$ *e.g.* with a neural network: specifically, the training loss considered in practice is

$$J_{\text{SM}}(\theta, \lambda) = \int_0^{T_1} \lambda(t) \mathbb{E}_{p_t} \left[ \| \nabla \log p_t - s_\theta(t, \cdot) \|^2 \right] dt, \tag{3.4}$$

for some strictly positive weight function $t \to \lambda(t)$. Notably, although the score $\nabla \log p_t$ is unknown, this loss can be estimated with standard score-matching techniques (Vincent, 2011; Song et al., 2020; 2021b).

When $\lambda(t) = g(t)^2$, which corresponds to the likelihood weighting of Song et al. (2021a), we simply write $J_{\mathrm{SM}}(\theta)$. The corresponding sampling algorithm simulates the process

$$\begin{cases} Y_0 \sim \pi, \\ dY_t = -f(T_1 - t, Y_t)\, dt + \frac{M(t)}{2} g(T_1 - t)^2 s_\theta(T_1 - t, Y_t)\, dt + \sqrt{M(t) - 1}\, g(T_1 - t)\, dB_t, \end{cases} \tag{3.5}$$

until time $T_1$, and it takes $Y_{T_1}$ as an approximate sample of $p$. This reverse SDE can be approximated via standard general-purpose numerical solvers, or by taking advantage of the additional knowledge of the approximated score function $s_\theta(t, \cdot) \approx \nabla \log p_t$: for example, the *predictor-corrector* methods of Song et al. (2021b) alternate one discretized step for the reverse process (3.5) with several steps of a score-based corrector algorithm, such as Langevin dynamics or Hamiltonian Monte Carlo.

For ease of exposition, we will focus on the popular *Ornstein–Uhlenbeck* (OU) forward process

$$\begin{cases} X_0 \sim p, \\ dX_t = -X_t\, dt + \sqrt{2}\, dB_t, \end{cases} \tag{3.6}$$

which corresponds to a method known as *Denoising Diffusion Probabilistic Models (DDPMs)* (Ho et al., 2020), and is also referred to as Variance Preserving SDE in Song et al. (2021b). Its Fokker–Planck equation reads

$$\begin{cases} p_0 = p, \\ \partial_t p_t + \nabla \cdot [p_t(-x - \nabla \log p_t(x))] = 0. \end{cases} \tag{3.7}$$

The standard Gaussian $\gamma$ is the limiting distribution of the OU process: more precisely, if $Z \sim \gamma$ is independent of $X_0$, then $X_t \sim e^{-t} X_0 + \sqrt{1 - e^{-2t}} Z$, and $p_t$ converges to $\gamma$ *e.g.* in Wasserstein distance $W_2$ and in relative entropy $D_{\mathrm{KL}}(\cdot \,\|\, \gamma)$ (Villani, 2003, Chap. 9). Restricting to the OU process (or its time reparametrization) is commonly done in the theoretical literature, see Lee et al. (2022); Bortoli (2022); Chen et al. (2023b; 2022a); as for these works, our techniques can be extended to other choices of the forward process, such as those considered in Song et al. (2020, Sec. 3.4).

**Notation.** We denote by $\gamma_{y,t}$ the density of a normal random variable in $\mathbb{R}^d$ with mean $y$ and variance $t I_d$, and for compactness we write $\gamma_t = \gamma_{0,t}$ and $\gamma = \gamma_1$. With abuse of notation, we identify the law of a random variable with the corresponding probability density. Given two probability measures $\mu, \nu$, the KL-divergence is defined by $D_{\mathrm{KL}}(\mu \,\|\, \nu) = \int \log\left(\frac{d\mu}{d\nu}\right) d\mu$ if $\mu$ is absolutely continuous with respect to $\nu$, and $D_{\mathrm{KL}}(\mu \,\|\, \nu) = +\infty$ otherwise; if $\mu, \nu$ have finite second moment, the 2-Wasserstein distance is defined by $W_2^2(\mu, \nu) = \inf_{X \sim \mu, Y \sim \nu} \mathbb{E}[\|X - Y\|^2]$. We denote by $\mathcal{P}(\mathbb{R}^d)$ the space of probability measures on $\mathbb{R}^d$. Throughout this paper, we denote by $p$ the target probability measure and by $p_t$ its law following the OU process; correspondingly, we consider random variables $X \sim p$, $X_t \sim p_t$. We use the symbol $\lesssim$ to denote an inequality up to an absolute positive multiplicative constant.

## 4 Improved Wasserstein-convergence in continuous time via prediction-correction

**Description of the algorithm.** We consider the following predictor-corrector algorithm. First, we run the OU forward process (3.6) until time $T_1$ and, for $0 \le t \le T_1$, we approximate $\nabla \log p_t(x)$ with $s_\theta(t, x)$. Next, we approximate $p_{T_1}$ by following an *inexact* Langevin dynamics started at $\gamma$ until time $T_2$:

$$\begin{cases} Z_0 \sim \gamma, \\ dZ_t = s_\theta(T_1, Z_t)\, dt + \sqrt{2}\, dB_t, \qquad 0 \le t \le T_2. \end{cases} \tag{4.1}$$

We remark that the Langevin dynamics (4.1) is *inexact* as it uses $s_\theta(T_1, x)$ in place of $\nabla \log p_{T_1}(x)$, since we have access to the former but not to the latter. The idea is that, as these two quantities are close, the random variable $Z_{T_2}$ provides an approximate sample of $p_{T_1}$, for sufficiently large $T_2$. Then, we approximate the original distribution $p$ by following a deterministic reverse process that starts from $Z_{T_2}$:

$$\begin{cases} Y_0 = Z_{T_2}, \\ dY_t = Y_t\, dt + s_\theta(T_1 - t, Y_t)\, dt, \qquad 0 \le t \le T_1. \end{cases} \tag{4.2}$$

We let $q_t = \text{law}(Y_t)$ for $t \in [0, T_1]$ and $\sigma_t = \text{law}(Z_t)$ for $t \in [0, T_2]$. In particular, we have $q_0 = \sigma_{T_2}$. Here, the *prediction-correction* consists in starting (4.2) from $Z_{T_2}$, instead of $\gamma$. We also note that this reverse process is deterministic except for its initialization $Y_0 = Z_{T_2}$. This allows to use standard numerical methods for solving ordinary differential equations, and in particularly exponential integrator schemes have shown remarkable performances (Lu et al., 2022). Finally, we take $Y_{T_1}$ to be an approximate sample from $p$. For stability reasons, we can also choose a small time $0 < \tau \ll T_1$ and stop the reverse process (4.2) at time $T_1 - \tau$, taking $Y_{T_1-\tau}$ as an approximate sample from $p$. This is commonly done in practice, see *e.g.* Song et al. (2021b, Sec. C).

**Assumptions.** Throughout this section, we consider the following assumptions.

(A1) The estimator $s_\theta \colon [0, T_1] \times \mathbb{R}^d \to \mathbb{R}^d$ is Lipschitz continuous. Moreover, for $t \in [0, T_1]$ we denote by $L_s(t) \in \mathbb{R}$ the *one-sided Lipschitz constant* for $s_\theta(t, \cdot)$, such that for all $x, y \in \mathbb{R}^d$,

$$(s_\theta(t, x) - s_\theta(t, y)) \cdot (x - y) \leq L_s(t)\|x - y\|^2.$$

(A2) $X \sim p$ is norm-subgaussian.

Condition (A1) is mild: in fact, $s_\theta$ is typically given by a neural network, which corresponds to a Lipschitz function for most practical activations. We emphasize that the requirement on the Lipschitz constant of $s_\theta$ is purely qualitative, in the sense that it does not enter our bounds (as opposed to the one-sided Lipschitz constant, which instead plays a quantitative role in the bounds).

As for condition (A2), we recall that an $\mathbb{R}^d$-valued random variable $X$ is norm-subgaussian if its euclidean norm $\|X\|$ is subgaussian (for details and a formal definition, see Appendix A). We denote by $\|X\|_{\text{SG}}$ the corresponding norm. Bounded random variables and subgaussian ones (in the sense of Vershynin (2018, Def. 3.4.1)) are norm-subgaussian, which covers most practical cases (e.g., in image generation pixels are usually rescaled in $[0, 1]$). Other properties of norm-subgaussian random vectors are established in Jin et al. (2019).

**Performance of the proposed algorithm.** Our main result is stated below.

**Theorem 4.1.** *Let Assumptions (A1)-(A2) hold, and let $p_t$ be obtained via the forward OU process in (3.6). Pick $0 < \delta < 1$, $T_2 > 0$, $T_1 \geq \frac{1}{2} \log\left(2 + 172 \frac{\|X\|_{\text{SG}}^2}{\delta} + \frac{d}{2\delta}\right)$ and a small early stopping time $0 < \tau \leq \min\left\{T_1, \frac{d}{2\|X\|_{\text{SG}}^2}\right\}$. Let $p_\theta = q_{T_1-\tau} = \text{law}(Y_{T_1-\tau})$, where $Y_{T_1-\tau}$ is obtained from the reverse process in (4.2). Consider the loss functions given by*

$$b(t) = \mathbb{E}_{p_t}\left[\|\nabla \log p_t - s_\theta(t, \cdot)\|^2\right], \quad \epsilon_{\text{MGF}} = \log \mathbb{E}_{p_{T_1}}\left[\exp\left(\frac{1}{1-\delta}\|\nabla \log p_{T_1} - s_\theta(T_1, \cdot)\|^2\right)\right].$$

*Then, the distance between the output $p_\theta$ and the target distribution $p$ can be bounded as follows:*

$$W_2(p, p_\theta) \leq \sqrt{3\tau d} + \int_\tau^{T_1} \sqrt{b(t)} I_\tau(t)\, dt + I_\tau(T_1)\sqrt{\frac{2}{1-\delta}\left(\delta e^{-\frac{(1-\delta)T_2}{2}} + 2\epsilon_{\text{MGF}}\right)} \tag{4.3}$$

$$\leq \sqrt{3\tau d} + \sqrt{\frac{J_{SM}(\theta)}{2} \int_\tau^{T_1} I_\tau(t)^2\, dt} + I_\tau(T_1)\sqrt{\frac{2}{1-\delta}\left(\delta e^{-\frac{(1-\delta)T_2}{2}} + 2\epsilon_{\text{MGF}}\right)}, \tag{4.4}$$

*where $I_\tau(t) = \exp\left(t - \tau + \int_\tau^t L_s(r)\, dr\right)$.*

The right-hand side of (4.3)–(4.4) consists of three error terms, due to *(i)* the early stopping of the reverse process (4.2), *(ii)* the approximation of the score function $s_\theta(t, \cdot) \approx \nabla \log p_t$ in (4.2), and *(iii)* the approximation $q_0 = \sigma_{T_2} \approx p_{T_1}$ from the output of the Langevin dynamics (4.1). A key feature of Theorem 4.1 is that, to have a vanishing sampling error, one does *not* need $T_1 \to \infty$. In fact, consider a sequence of estimators

satisfying the additional technical assumption[3] $\limsup_{J_{\mathrm{SM}}\to 0}\int_\tau^{T_1}\exp\left(2\int_\tau^t L_s(r)\,dr\right)dt < \infty$. Then, letting $\tau \to 0$, $J_{\mathrm{SM}} \to 0$, $\epsilon_{\mathrm{MGF}} \to 0$ and $T_2 \to \infty$, with $T_1$ fixed, the convergence of $p_\theta$ to $p$ in $W_2$ distance follows from Theorem 4.1. We now discuss the roles of $T_1, T_2, \tau$ and of the free parameter $\delta$.

The role of $T_1$ is to ensure the regularity of $p_{T_1}$. Specifically, we show that $p_{T_1}$ satisfies a log-Sobolev inequality with constant at least $1 - \delta$, which leads to the mild logarithmic dependence of $T_1$ on the subgaussian norm $\|X\|_{\mathrm{SG}}$. In fact, the dependence on $d$ can be even dropped (although the subgaussian norm $\|X\|_{\mathrm{SG}}$ may still depend on $d$): if $T_1 \geq \frac{1}{2}\log\left(2 + 172\frac{\|X\|_{\mathrm{SG}}^2}{\delta}\right)$, then (4.3) and (4.4) hold with $\frac{d}{3}\exp\left(-\frac{(1-\delta)T_2}{2}\right)$ in place of $\delta\exp\left(-\frac{(1-\delta)T_2}{2}\right)$ (see the last term of the expression). Choosing $T_1 > 0$ is necessary, as performing Langevin dynamics directly for $p$ works poorly. This was already observed in Song & Ermon (2019) and served precisely as a motivation for SGMs.

The role of $T_2$ is to improve the accuracy of sampling from $p_{T_1}$ and, due to the regularity of this distribution, the convergence is exponential in $T_2$. Taking $T_2$ large, instead of $T_1$, is beneficial, as the neural network needs to approximate the score of the forward process only until time $T_1$ and, correspondingly, $J_{\mathrm{SM}}$ increases with $T_1$. In contrast, Kwon et al. (2022, Thm. 1, Cor. 2) requires $T_1 \to \infty$ to achieve convergence, and a full quantitative analysis of the dependence of the bounds on $T_1$ is missing.

The role of $\tau$ is to allow for early stopping in the reverse process (4.2). If that is not needed (since *e.g.* the distribution $p$ is sufficiently regular), then one can simply take the limit $\tau \to 0$ in (4.3)-(4.4).[4]

The role of $\delta$ is to provide a trade-off between $T_1$ and $T_2$. We remark that the result of Theorem 4.1 holds for any $\delta \in (0, 1)$, and smaller values of $\delta$ give a faster decay of the error term coming from the Langevin dynamics, due to the larger log-Sobolev constant of $p_{T_1}$. This improvement comes at the price of a tighter lower bound for $T_1$.

We highlight that our convergence result does not need the condition $\lim_{t\to\infty} L_s(t) = -1$, which was introduced in Kwon et al. (2022). This requirement was heuristically justified by the one-sided Lipschitz constant of the stationary distribution $\gamma$ of the forward process (3.6) being $-1$, but obtaining a rigorous control on $\lim_{t\to\infty} L_s(t)$ only from the $L^2$ loss of the score estimation remained challenging. To circumvent this issue, we instead introduce the additional loss $\epsilon_{\mathrm{MGF}}$ in Theorem 4.1, which concerns the score estimation *only* at time $T_1$. By Jensen's inequality, this is a stronger loss than the usual $L^2$ one ($\epsilon_{\mathrm{MGF}} \geq \frac{1}{1-\delta}b(T_1)$). However, a simple truncation of the estimator $s_\theta(T_1, \cdot)$ allows us to control $\epsilon_{\mathrm{MGF}}$ in terms of $b(T_1)$. This is formalized in the result below.

**Theorem 4.2.** *Let Assumptions (A1)-(A2) hold, and let $p_t$ be obtained via the forward OU process in* (3.6) *starting from $X \sim p$. Pick $0 < \delta < 1$ and $T_1 \geq \log\left(\frac{16}{\delta}d(\|X\|_{\mathrm{SG}} + 1)\right)$. Define the estimator $\widetilde{s}_\theta \colon \mathbb{R}^d \to \mathbb{R}^d$ by*

$$[\widetilde{s}_\theta(x)]_i = \begin{cases} [s_\theta(T_1, x)]_i, & \text{if } |-x_i - [s_\theta(T_1, x)]_i| \leq \ell, \\ -x_i - \ell, & \text{if } [s_\theta(T_1, x)]_i < -x_i - \ell, \\ -x_i + \ell, & \text{if } [s_\theta(T_1, x)]_i > -x_i + \ell, \end{cases} \tag{4.5}$$

*where $\ell = \ell(x) = \frac{\delta}{2d}(1 + \|x\|)$. Fix $0 < \epsilon < 1$. Then, for all $0 < \beta \leq \frac{1}{36\delta^2}$, we have*

$$\log \mathbb{E}_{p_{T_1}}\left[\exp\left(\beta\|\nabla\log p_{T_1} - \widetilde{s}_\theta\|^2\right)\right] \leq \epsilon, \tag{4.6}$$

*provided that*

$$b(T_1) \leq \frac{1}{34\beta}\epsilon^{1+36\beta\delta^2}. \tag{4.7}$$

---

[3]This condition was implicitly needed in Kwon et al. (2022) for the same reasons. To see that it is reasonable, notice that $L_s$ is upper bounded by the Lipschitz constant $\mathrm{Lip}(s_\theta)$, which is expected to be similar to $\mathrm{Lip}(\nabla\log p_t)$ as $J_{\mathrm{SM}} \to 0$. The latter is well behaved by the regularization properties of the OU flow: if $p$ has bounded support, then $\int_\tau^{T_1}\mathrm{Lip}(\nabla\log p_t)\,dt < \infty$ for all $0 < \tau < T_1$ (Chen et al., 2023b, Lem. 20). Note also that the one-sided Lipschitz constant can be negative (e.g., for $\gamma_t$ it is equal to $-\frac{1}{t}$), which helps with the convergence of the integral.

[4]This passage can be justified by an application of monotone convergence, after estimating the one-sided Lipschitz constant $L_s(t)$ with the Lipschitz constant of $s_\theta(t, \cdot)$.

A combination of Theorems 4.1 and 4.2 gives an end-to-end convergence result in $W_2$ distance requiring only a control on the $L^2$ loss $\{b(t)\}_{t\in[0,T_1]}$, which is the object minimized in practice. We regard the truncation of the estimator $s_\theta(T_1, \cdot)$ carried out in Theorem 4.2 as purely technical. In fact, even if the estimator $s_\theta(T_1, \cdot)$ explicitly minimizes the training loss $b(T_1)$, one also expects the stronger loss $\epsilon_{\text{MGF}}$ to be small, when $\delta$ is sufficiently small. This is confirmed by the numerical results of Figure 1, for which we do not replace the estimator $s_\theta(T_1, \cdot)$ with $\widetilde{s}_\theta$. Specifically, the plots show that, having fixed $T_1$, both $W_2(p, p_\theta)$ (in orange) and $W_2(p_{T_1}, q_0)$ (in blue) decrease as a function of the running time $T_2$ of the inexact Langevin dynamics (4.1) for different standard datasets.

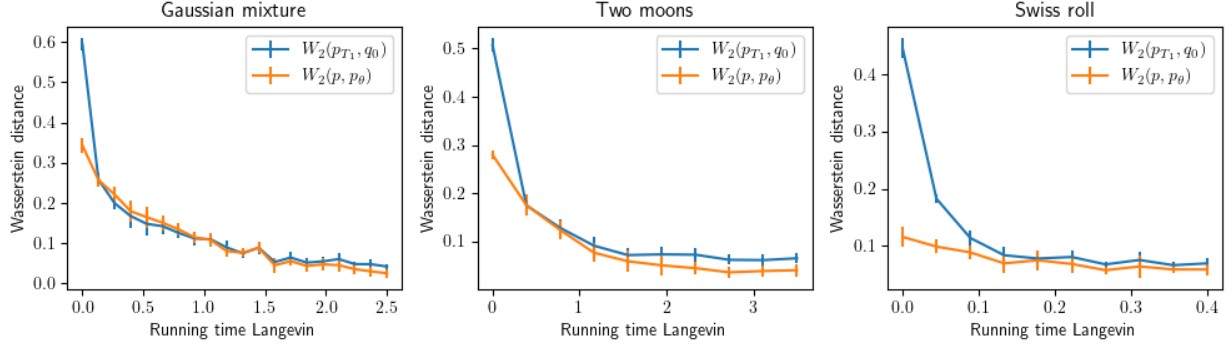

Figure 1: Simulation results for an asymmetric mixture of two gaussians (left), the two moons dataset (center) and the rescaled swiss roll (right). For fixed $T_1$ and variable $T_2$, we plot in blue the $W_2$ distance between the perturbed measure $p_{T_1}$ and the output of (4.1), while we plot in orange the $W_2$ distance between the true distribution $p$ and the output of the algorithm $p_\theta$. As expected, both quickly decrease as $T_2$ increases.

### 4.1 Proof ideas

**Analysis of the reverse process** (4.2). To obtain Theorem 4.1, we start with the analysis of the reverse process (4.2). By adapting the argument in Kwon et al. (2022), we derive the following bound on the Wasserstein distance between $p_\tau$ and $q_{T_1-\tau}$.

**Proposition 4.3.** *Under Assumptions (A1)-(A2), for $0 \le \tau < T_1$, we have*

$$W_2(p_\tau, q_{T_1-\tau}) \le I_\tau(T_1) W_2(p_{T_1}, q_0) + \int_\tau^{T_1} I_\tau(t)\sqrt{b(t)}\, dt. \tag{4.8}$$

For completeness we include a proof in Appendix B, since compared to Kwon et al. (2022) we consider $M(t) \equiv 1$ (and not $M(t) \equiv 2$), a different starting distribution for $Y_0$, and early stopping. In addition, we need the following short-time estimate on $W_2(p, p_\tau)$, which is proved in Appendix B.

**Lemma 4.4.** *Suppose that $M := \int_{\mathbb{R}^d} \|x\|^2 p(x)dx < \infty$. Then, for $0 < \tau < \frac{d}{M^2}$, we have*

$$W_2(p, p_\tau) \le \sqrt{3\tau d}.$$

Using the triangle inequality for $W_2$, the combination of Proposition 4.3 and Lemma 4.4 readily gives

$$W_2(p, q_{T_1-\tau}) \le \sqrt{3\tau d} + I_\tau(T_1) W_2(p_{T_1}, q_0) + \int_\tau^{T_1} \sqrt{b(t)} I_\tau(t)\, dt. \tag{4.9}$$

This bound shows that to achieve convergence, we need *(i)* small $\tau$, *(ii)* $b(t) \to 0$ (which is reasonable, since $s_\theta$ is obtained by minimizing the $L^2$ loss), and *(iii)* $W_2(q_0, p_{T_1}) \to 0$. The latter condition corresponds to choosing a good starting distribution for the reverse process (instead of $\gamma$), and it is ensured by the inexact Langevin dynamics (4.1), which will be analyzed next.

**Analysis of inexact Langevin dynamics** (4.1). Recall the definition of the log-Sobolev inequality.

**Definition 4.5.** *A probability measure $\mu$ satisfies a log-Sobolev inequality with constant $\kappa > 0$ (notation: $\mathrm{LSI}(\kappa)$) if, for all probability measures $\nu$,*

$$D_{KL}(\nu \,\|\, \mu) \leq \frac{1}{2\kappa} \mathcal{I}_\mu(\nu), \tag{4.10}$$

*where $\mathcal{I}_\mu(\nu)$ is the relative Fisher Information, defined by*

$$\mathcal{I}_\mu(\nu) = \begin{cases} \int_{\mathbb{R}^d} \left\| \nabla \log \frac{d\nu}{d\mu} \right\|^2 d\nu = 4 \int_{\mathbb{R}^d} \left\| \nabla \sqrt{\frac{d\nu}{d\mu}} \right\|^2 d\mu, & \text{if } \nu \ll \mu \text{ and } \sqrt{\frac{d\nu}{d\mu}} \in H^1(\mu), \\ +\infty, & \text{otherwise.} \end{cases} \tag{4.11}$$

*Here, $H^1(\mu)$ denotes the weighted Sobolev space. If $\mu$ satisfies a log-Sobolev inequality, we use the notation $C_{\mathrm{LS}}(\mu)$ for its optimal constant.*

**Remark 4.6.** *If $\mu$ satisfies $\mathrm{LSI}(\kappa)$ for some $\kappa > 0$, then it also satisfies the transport-entropy inequality (Otto & Villani, 2000) (see also Gozlan & Léonard (2010, Thm. 8.12) for an overview of related results)*

$$W_2(\nu, \mu) \leq \sqrt{\frac{2}{\kappa} D_{KL}(\nu \,\|\, \mu)}. \tag{4.12}$$

It is well known that Langevin dynamics for a measure $\mu$ converges exponentially fast in KL-divergence if $\mu$ satisfies a log-Sobolev inequality, see *e.g.* Villani (2003); Vempala & Wibisono (2019). A similar convergence result has recently been proved in Yingxi Yang & Wibisono (2023) for the inexact Langevin dynamics

$$\begin{cases} Z_0 \sim \nu, \\ dZ_t = s_\theta(Z_t)\, dt + \sqrt{2}\, dB_t, \end{cases} \tag{4.13}$$

where $s_\theta$ approximates the score $\nabla \log \mu$. For convenience, we state Theorem 1 of Yingxi Yang & Wibisono (2023) below.

**Theorem 4.7.** *Let $\mu, \nu$ be probability measures with full support that admit densities with respect to the Lebesgue measure. Suppose that $\mu$ satisfies a log-Sobolev inequality and let $\kappa = C_{\mathrm{LS}}(\mu)$. Then, the time-marginal law $\nu_t$ of the inexact Langevin dynamics (4.13) satisfies, for $t \geq 0$,*

$$D_{KL}(\nu_t \,\|\, \mu) \leq e^{-\frac{1}{2}\kappa t} D_{KL}(\nu \,\|\, \mu) + 2 \log \mathbb{E}_\mu \left[ \exp\left( \frac{1}{\kappa} \| \nabla \log \mu - s_\theta \|^2 \right) \right].$$

The application of Theorem 4.7 requires $p_{T_1}$ to satisfy a log-Sobolev inequality, as well as the estimation of $C_{\mathrm{LS}}(p_{T_1})$. To ensure this, we notice that $p_{T_1} = \mathrm{law}\left( e^{-T_1} X + \sqrt{1 - e^{-2T_1}} Z \right)$ where $Z \sim \gamma$ is an independent Gaussian, and apply recent results from Chen et al. (2021a) that quantify the log-Sobolev constant of the convolution of a subgaussian probability measure with a Gaussian having sufficiently high variance (cf. Lemma B.3 and Theorem B.4 in Appendix B). In this way, we deduce that $C_{\mathrm{LS}}(p_{T_1}) \geq 1 - \delta$. Next, in order to apply Theorem 4.7 with $\mu = p_{T_1}$ and $\nu = \gamma$, we need an estimate of $D_{KL}(\gamma \,\|\, p_{T_1})$. This is provided by the result below, which is proved in Appendix B.

**Lemma 4.8.** *Let $p \in \mathcal{P}(\mathbb{R}^d)$ with $M := \int_{\mathbb{R}^d} \|x\|^2 \, dp(x) < \infty$. Then, for $t > 0$ we have*

$$D_{KL}(\gamma \,\|\, p_t) \leq \frac{d}{2\sigma_t} \left( \frac{M}{d} e^{-2t} + \sigma_t \log \sigma_t - \sigma_t + 1 \right),$$

*with $\sigma_t = 1 - e^{-2t}$. Thus, for $t \geq \max\left( \log(\sqrt{2}), \log\left( \sqrt{(M + d/2)/\delta} \right) \right)$, we have $D_{KL}(\gamma \,\|\, p_t) \leq \delta$, while for $t \geq \max\left( \log(\sqrt{2}), \log\left( \sqrt{3M} \right) \right)$ we have $D_{KL}(\gamma \,\|\, p_t) \leq \frac{d}{3}$.*

By combining the estimate on $D_{\mathrm{KL}}(\sigma_{T_2} \,\|\, p_{T_1})$ given by Theorem 4.7 with the transport-entropy inequality (4.12) (which also uses $C_{\mathrm{LS}}(p_{T_1}) \geq 1 - \delta > 0$) and the above lemma, we obtain an upper bound on $W_2(\sigma_{T_2}, p_{T_1}) = W_2(q_0, p_{T_1})$. Combining this upper bound with (4.9) gives the desired inequality (4.3). The inequality (4.4) then follows from the Cauchy-Schwarz inequality, which concludes the proof of Theorem 4.1. The complete argument is contained in Appendix B.

**Controlling a stronger loss (Theorem 4.2).** It is not difficult to construct an estimator $s_\theta(T_1, \cdot)$ such that $b(T_1)$ is arbitrarily small and $\epsilon_{\mathrm{MGF}}$ diverges. In fact, consider estimating the score of the standard Gaussian $\gamma$, given by $\nabla \log \gamma(x) = -x$, via a sequence of estimators of the form $s_M(x) = -x \mathbb{1}_{\|x\| \leq M}$. Then, we have $\lim_{M \to \infty} b(T_1) = 0$, but $\epsilon_{\mathrm{MGF}}$ is infinite for all $M \geq 0$. This might seem discouraging, but we can prevent such pathological problems by leveraging our knowledge about the target score function. This allows us to fix predictions of the estimator $s_\theta(T_1, x)$ that happen to be very far from the target value $\nabla \log p_{T_1}(x)$.

More precisely, by choosing $T_1$ according to the prescription of Theorem 4.2, we can ensure that $\nabla \log p_{T_1}(x)$ lies in a region around the score of the standard Gaussian $\nabla \log \gamma = -x$, i.e.,

$$|-x_i - \partial_i \log p_{T_1}| \leq \frac{\delta}{2d}(1 + \|x\|), \quad \text{for all } i \in \{1, \dots, d\},$$

see Lemma C.2. This is illustrated in Figure 2, in one dimension: the green dashed line represents $\nabla \log \gamma = -x$, and our choice of $T_1$ guarantees that $\nabla \log p_{T_1}$ lies in the region delimited by the blue and purple lines. Whenever $s_\theta$ returns a value outside this region, we correct it by choosing the closest value on the boundary. This leads to the definition of the estimator $\widetilde{s}_\theta$ given by (4.5). For this new estimator, we can now convert an $L^2$ error into

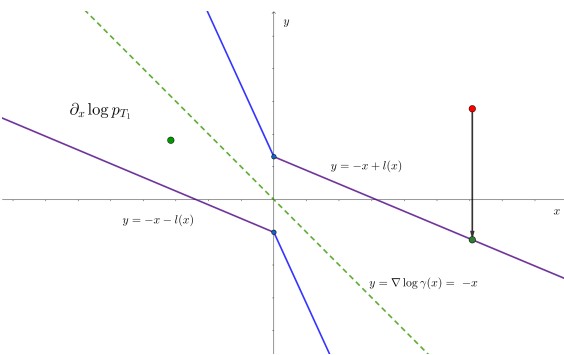

Figure 2: The confinement region for $\nabla \log p_{T_1}$.

an upper bound for $\epsilon_{\mathrm{MGF}}$, which gives Theorem 4.2. The argument crucially relies on the fast decay of the tails of $p_{T_1}$, thanks to the similarity with the standard Gaussian $\gamma$, and exploits the confinement knowledge to deduce a converse bound to Jensen's inequality, cf. Lemma A.4. The complete proof is contained in Appendix C.

## 5 Convergence of discretized schemes in total variation distance

In practical implementations, the continuous time dynamics needs to be approximated by a discrete time scheme. The existing literature for discretizations of the reverse process (Chen et al., 2023b; 2022a; Lee et al., 2022) focuses on the reverse SDE instead of the ODE (*i.e.*, $M(t) = 2$ instead of $M(t) = 1$ in (3.3)), and on information-divergence metrics instead of the Wasserstein distance, as for the latter the analysis seems significantly more complicated (Chen et al., 2023b, Sec. 4). The two-stage algorithm (with a stochastic reverse process) considered in this paper is compatible with such recent analyses: we illustrate this in the present section by providing convergence guarantees for a natural discretization of the algorithm. The argument proceeds in a similar way to Section 4. In particular, a key role is played again by Lemmas B.3 and 4.8: the first guarantees that $p_{T_1}$ satisfies a log-Sobolev inequality with a good constant, so that a discretization of the inexact Langevin dynamics performs well, while the second ensures that the Gaussian distribution $\gamma$ is a good initialization for the algorithm.

For the discretization of (4.1), we consider following the inexact Langevin algorithm with variable step sizes $h_k > 0$:

$$\begin{cases} Z_0 \sim \gamma, \\ Z_{k+1} = Z_k + h_k s_\theta(T_1, Z_k) + \sqrt{2h_k} B_k, \end{cases} \tag{5.1}$$

where $(B_k)_k \overset{\text{iid}}{\sim} \mathcal{N}(0, I_d)$. This is run for a variable number of steps $N_2$, and we now denote by $\sigma_k$ the law of $Z_k$.

Convergence guarantees for $Z_k$ as $k \to \infty$ can be obtained from the analysis in Yingxi Yang & Wibisono (2023, Thm. 2), which we slightly modify to take into account a decaying step size: this will give a logarithmic improvement on the computational complexity, in the same spirit as Dalalyan & Karagulyan (2019). As in Section 4 (and as in Yingxi Yang & Wibisono (2023)), the analysis of the Langevin algorithm introduces a modified loss $\tilde{\epsilon}_{\mathrm{MGF}}$, which is stronger than the standard $L^2$ error on the accuracy of the score estimate. However, at time $T_1$ this loss can be controlled again thanks to Theorem 4.2 (cf. also Remark 5.2).

As for the reverse process, to take advantage of the existing literature, we consider a discretization of the reverse SDE instead of the reverse ODE (corresponding to $M(t) = 2$ instead of $M(t) = 1$ in (3.5)). The chosen numerical method is the popular exponential integrator scheme (Zhang & Chen, 2023), so that the second stage of the algorithm is given by

$$
\begin{cases}
Y_0 = Z_{N_2}, \\
Y_{k+1} = Y_k e^{\tilde{h}_k} + 2s_\theta(t_k, Y_k)\left(e^{\tilde{h}_k} - 1\right) + \sqrt{e^{2\tilde{h}_k} - 1}\,\tilde{B}_k, \\
t_k = T_1 - \sum_{i=0}^{k-1} \tilde{h}_i,
\end{cases}
\tag{5.2}
$$

for a sequence of step sizes $(\tilde{h}_k)_{k=0}^{N_1-1}$ such that $\sum_{k=0}^{N_1-1} \tilde{h}_k \le T_1$ and for $(\tilde{B}_k)_k \overset{\text{iid}}{\sim} \mathcal{N}(0, I_d)$. The output $Y_{N_1}$ is finally taken as an approximate sample from $p$, and we denote now by $p_\theta$ its law; under appropriate assumptions, we derive convergence guarantees of $p_\theta$ to $p$ in total variation distance. In place of the integrated $L^2$ loss in (3.4), for discrete time schemes it is natural to introduce the analogous loss

$$
\widehat{J_{\text{SM}}} = \sum_{k=0}^{N_1-1} \tilde{h}_k \mathbb{E}_{p_{t_k}}\left[\|\nabla \log p_{t_k} - s_\theta(t_k, \cdot)\|^2\right].
$$

We can think of $\widehat{J_{\text{SM}}}$ as an approximation of $J_{\text{SM}}$: it is a standard and realistic assumption that $\widehat{J_{\text{SM}}}$ can be made arbitrarily small with sufficient data and model capacity. The errors arising in the reverse process (5.2) (due to the inaccuracy of the score estimate and the use of the discrete scheme) can be bounded thanks to the recent theoretical literature on the performance of diffusion models, see e.g. Chen et al. (2023b; 2022a); together with the analysis of (5.1), this allows us to deduce end-to-end convergence guarantees for the two-stage algorithm.

To illustrate this, we present below one such result which exploits the analysis of Chen et al. (2022a, Thm. 1), and we refer the reader to Appendix D for the proof. Additional results under different assumptions and choices of the step size can be deduced by adapting other arguments (e.g. Chen et al. (2022a, Thm. 2)). Specifically, we consider a constant step size for the reverse process and a standard smoothness condition (Chen et al., 2023b; 2022a).

**Theorem 5.1.** *Under Assumption (A1), pick $0 < \delta \le \frac{1}{2}$ and $T_1 \ge \frac{1}{2}\log\left(2 + 172\frac{\|X\|_{\text{SG}}^2}{\delta} + \frac{d}{2\delta}\right)$. Suppose in addition that $\nabla \log p_t$ is $L_1$-Lipschitz for $t \in [0, T_1]$ and that $s_\theta(T_1, \cdot)$ is $L_2$-Lipschitz with $L_1, L_2 \ge 1$, and consider the modified loss at time $T_1$*

$$
\tilde{\epsilon}_{\text{MGF}} := \log \mathbb{E}_{p_{T_1}}\left[\exp\left(\frac{9}{1-\delta}\|\nabla \log p_{T_1} - s_\theta(T_1, \cdot)\|^2\right)\right].
\tag{5.3}
$$

*Then, for the algorithm described above with step sizes $h_k = \frac{1}{24L_1 L_2 + \frac{k+1}{16}}$ and $\tilde{h}_k = \tilde{h} = \frac{T_1}{N_1} \le 1$, we have that*

$$
\begin{aligned}
TV(p, p_\theta) &\lesssim \sqrt{\widehat{J_{SM}} + \frac{dL_1^2 T_1^2}{N_1}} + \sqrt{\left(\frac{L_1 L_2}{N_2 + 1}\right)^2 + \frac{dL_2^2}{N_2 + 1} + \tilde{\epsilon}_{\text{MGF}}} \\
&= \sqrt{\widehat{J_{SM}} + dL_1^2 T_1 \tilde{h}} + \sqrt{\left(\frac{L_1 L_2}{N_2 + 1}\right)^2 + \frac{dL_2^2}{N_2 + 1} + \tilde{\epsilon}_{\text{MGF}}}.
\end{aligned}
\tag{5.4}
$$

**Remark 5.2.** *To deduce convergence from the above result assuming only $L^2$ accuracy of the score, we need to control $\tilde{\epsilon}_{\text{MGF}}$. This can be done again using Theorem 4.2, where we now choose $\beta = \frac{9}{1-\delta}$ and take $0 < \delta < 0.054$ to fulfill $\beta \le \frac{1}{36\delta^2}$.*

**Remark 5.3** (Complexity of sampling). *Suppose that the goal is to achieve $TV(p, p_\theta) \le \epsilon$ for some $0 < \epsilon < 1$. A typical assumption is to be able to control the $L^2$ error of the score approximation; hence, by the remark above, we can assume that $\widehat{J_{SM}}, \tilde{\epsilon}_{\text{MGF}} \lesssim \epsilon^2$, as needed in the bound (5.4). Consequently, (5.4) shows that the algorithm needs at most $N$ steps to ensure $TV(p, p_\theta) \le \epsilon$, with*

$$
N = N_1 + N_2 \lesssim \frac{d}{\epsilon^2} \cdot \left(L_1^2 \log^2(d + \|X\|_{\text{SG}}) + L_2^2\right).
$$

**Remark 5.4.** *If we choose also for the inexact Langevin algorithm* (5.1) *a fixed step size* $h_k = h \leq \frac{1}{24 L_1 L_2}$, *we obtain instead the bound*

$$TV(p, p_\theta) \lesssim \sqrt{\widehat{J_{SM}} + \frac{d L_1^2 T_1^2}{N_1}} + \sqrt{e^{-\frac{1}{8} h N_2} + L_2(L_1 + L_2) dh + \tilde{\epsilon}_{\mathrm{MGF}}}.$$

*In this case, for the number of steps* $N$ *to achieve* $TV(p, p_\theta) \leq \epsilon$, *we have that*

$$N = N_1 + N_2 \lesssim \frac{d}{\epsilon^2} \cdot \left( L_1^2 \log^2(d + \|X\|_{\mathrm{SG}}) + L_2(L_1 + L_2) \log \frac{1}{\epsilon} \right).$$

## 5.1 Comparison with previous results

To highlight a few favorable properties of the predictor-corrector algorithm, we compare the result above with Chen et al. (2022a, Thm. 1), translated into total variation distance via Pinsker's inequality. The authors of Chen et al. (2022a) consider, under similar assumptions, the standard sampling scheme based on a discretization of the reverse SDE without the prior Langevin algorithm or any corrector; in other words, they consider (5.2) with constant step size initialized at $Z_0 \sim \gamma$. For the corresponding output distribution $\widehat{p_\theta}$ they prove the bound

$$\mathrm{TV}(p, \widehat{p_\theta}) \lesssim \sqrt{(M + d) e^{-2T_1} + \widehat{J_{\mathrm{SM}}} + \frac{d L^2 T_1^2}{N_1}}$$

$$= \sqrt{(M + d) e^{-2T_1} + \widehat{J_{\mathrm{SM}}} + d L^2 T_1 \tilde{h}}, \tag{5.5}$$

where $M$ is the second moment of $p$. Therefore, to achieve $\mathrm{TV}(p, \widehat{p_\theta}) \leq \epsilon$ the reverse SDE needs at most $N$ steps with

$$N \lesssim \frac{d L_1^2}{\epsilon^2} \log^2 \left( \frac{M + d}{\epsilon^2} \right).$$

Comparing the bounds (5.4) and (5.5) shows some advantages of the predictor-corrector schemes, arising from a fixed choice of $T_1$ independent of the desired sampling accuracy $\epsilon$.

1. The convergence result in (5.4), unlike (5.5), is stable with $T_1$. In other words, for a fixed choice of step size $\tilde{h}$ in the reverse process, the bound in (5.5) explodes as $T_1 \to \infty$, which is however necessary to minimize the error $(M + d) e^{-2T_1}$ arising from the approximation $p_{T_1} \approx \gamma$. Thus, there is a trade-off between the choice of $T_1$ and the step size in the reverse process. In contrast, this problem does not occur for the bound in (5.4), since $T_1$ is now fixed. For the specified choice of step sizes $h_k$ (which is independent of the desired accuracy), the error goes to 0 as $\tilde{h} \to 0$ and $N_2 \to \infty$. At the same time, the bound is now stable for any choice of the variable quantities $\tilde{h} \leq 1$ and $N_2 \geq 1$. Yingxi Yang & Wibisono (2023) is also aimed at obtaining stable convergence. However, the results therein apply only to distributions satisfying a log-Sobolev inequality and under stronger assumptions on the accuracy of the score.

2. To achieve convergence, the bound (5.5) requires learning the score $\nabla \log p_t$ on a time interval which increases as $T_1 \to \infty$: correspondingly, the error term $\widehat{J_{\mathrm{SM}}}$ increases too with $T_1$. For example, assuming that we have $L^2$-accuracy of $\epsilon^2$ at every time (i.e., $b(t_k) \leq \epsilon^2$ for every $t_k$, cf. Assumption 3 of Chen et al. (2023b)) we have that $\widehat{J_{\mathrm{SM}}} = \epsilon^2 T_1$, which diverges as $T_1 \to \infty$ with fixed $\epsilon > 0$ (see also Chen et al. (2023b, Thm. 2)). These problems do not occur with the bound in (5.4), since $T_1$ is fixed; this simplifies the training procedure of the neural network learning the score and contributes to the stability of the convergence result. At the same time, this further pushes the observation by Chen et al. (2023b) that "sampling is as easy as learning the score", in the sense that knowledge of the score $\nabla \log p_t$ for all times $t$ suffices for efficient sampling. Indeed, Theorem 5.1 shows the stronger result that, for norm-subgaussian distributions, knowledge of the score *on a fixed finite time interval* is actually enough.

3. Finally, when looking at the dependence on the desired accuracy $\epsilon$, the bound of (5.4) removes a factor of $\log^2 \left( \frac{1}{\epsilon} \right)$ in the number of steps required by the algorithm.

# 6    Concluding remarks

In this work, we give convergence guarantees for a variant of the popular predictor-corrector approach in the context of score-based generative modeling. Our analysis provides bounds that *(i)* require running the forward process only for a fixed time $T_1$, which does not depend on the final sampling accuracy, *(ii)* make minimal assumptions on the data (subgaussianity of the norm), *(iii)* exhibit a mild logarithmic dependence on the input dimension and on the tails of the data distribution, and *(iv)* allow for realistic assumptions on the score estimation, in the form of a control on the standard $L^2$ loss integrated over the finite time $T_1$.

**Acknowledgement**

Francesco Pedrotti and Jan Maas acknowledge support by the Austrian Science Fund (FWF) project 10.55776/F65. Marco Mondelli acknowledges support by the 2019 Lopez-Loreta prize.

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

# A  Additional and auxiliary lemmas

## Notation

Recall that a scalar random variable $X$ is *subgaussian* if there exists a constant $K > 0$ such that $\mathbb{E}\left[e^{\frac{X^2}{K^2}}\right] \leq 2$. Its subgaussian norm is defined by $\|X\|_{\psi_2} := \inf\left\{t > 0 : \mathbb{E}\left[e^{\frac{X^2}{t^2}}\right] \leq 2\right\}$. Furthermore, an $\mathbb{R}^d$-valued random variable $X$ is said to be norm-subgaussian if its euclidean norm $\|X\|$ is subgaussian. We define $\|X\|_{\mathrm{SG}} := \left\|\|X\|\right\|_{\psi_2}$ Note that both $\|\cdot\|_{\psi_2}$ and $\|\cdot\|_{\mathrm{SG}}$ are norms and, if $\mathrm{Supp}\, X \subset B(0, R)$ for some radius $R > 0$, then $\|X\|_{\mathrm{SG}} \leq \frac{R}{\sqrt{\log(2)}}$.

For the convenience of the reader, in the table below we recall the relevant notation used in the paper.

| Notation | Meaning |
|---|---|
| $\|X\|_{\psi_2}$ | Subgaussian norm of random variable |
| $\|X\|_{\mathrm{SG}} = \left\|\|X\|\right\|_{\psi_2}$ | Subgaussian norm of random vector |
| $p$ | Target distribution |
| $p_t$ | Perturbed distribution at time $t$ |
| $p_\theta$ | Output distribution of the algorithm |
| $s_\theta(t, x) \approx \nabla \log p_t(x)$ | Estimator for the score function |
| $L_s(t)$ | One-sided Lipschitz constant of $s_\theta(t, \cdot)$ |
| $\tau \geq 0$ | Early stopping time |
| $T_1 > 0$ | Running time of the forward process (OU) |
| $N_1 > 0$ | Number of steps for simulated reverse process |
| $T_2 \geq 0$ | Running time of inexact Langevin dynamics |
| $N_2 \geq 0$ | Number of steps for Langevin algorithm |
| $b(t) = \mathbb{E}_{p_t}\left[\|\nabla \log p_t - s_\theta(t, \cdot)\|^2\right]$ | $L^2$-error for the score approximation |
| $\epsilon_{\mathrm{MGF}} = \log \mathbb{E}_{p_{T_1}}\left[\exp\left(\frac{1}{1-\delta}\|\nabla \log p_{T_1} - s_\theta(T_1, \cdot)\|^2\right)\right]$ $\tilde{\epsilon}_{\mathrm{MGF}} = \log \mathbb{E}_{p_{T_1}}\left[\exp\left(\frac{9}{1-\delta}\|\nabla \log p_{T_1} - s_\theta(T_1, \cdot)\|^2\right)\right]$ | Stronger losses for the score at time $T_1$ |

## Auxiliary lemmas

First of all, we recall the following classical properties of the Gaussian distribution.

**Lemma A.1.** *Let $Z \sim \gamma_t$. Then,*
$$\|Z\|_{\mathrm{SG}} \leq 2\sqrt{dt}.$$
*Moreover, $C_{\mathrm{LS}}(\gamma_{x,t}) \geq \frac{1}{t}$ for all $x \in \mathbb{R}^d$ and $t > 0$.*

*Proof.* Without loss of generality, suppose that $Z \sim \gamma$ (i.e. $t = 1$). Recalling the moment generating function of the $\chi^2$-distribution, we have that, for $0 \leq c \leq \frac{1}{4d} < \frac{1}{2}$,

$$\mathbb{E}\left[e^{c\|Z\|^2}\right] = (1 - 2c)^{-\frac{d}{2}} \leq \frac{1}{1 - 2cd} \leq 2,$$

which proves the first claim.

The statement about the log-Sobolev constant is well known (it follows for example from the Bakry–Émery criterion, cf. Villani (2003, Thm. 9.9),Bakry et al. (2014)). □

**Lemma A.2.** *For all $t, c > 0$, we have*

$$\int_t^\infty e^{-cx^2} dx \leq \frac{1}{2ct} e^{-ct^2}.$$

*Proof.* As in Vershynin (2018, Prop 2.1.2), we have

$$\int_t^\infty e^{-cx^2} dx = \int_0^\infty e^{-c(x^2+2xt+t^2)} dx \leq e^{-ct^2} \int_0^\infty e^{-2ctx} dx = \frac{1}{2ct} e^{-ct^2}.$$

$\square$

The next lemma provides some useful estimates for norm-subgaussian random vectors (cf. Vershynin (2018); Jin et al. (2019)).

**Lemma A.3.** *Let $X \sim p$ be a norm-subgaussian random vector. The following hold:*

(*i*) *For all $s \geq 0$,*

$$\mathbb{P}(\|X\| \geq s) \leq 2e^{-\frac{s^2}{\|X\|_{\mathrm{SG}}^2}}. \tag{A.1}$$

*Moreover,*

$$\mathbb{E}\left[\|X\|^2\right] \leq 2\|X\|_{\mathrm{SG}}^2. \tag{A.2}$$

(*ii*) *For any $L > 0$ and $0 < c < \frac{1}{\|X\|_{\mathrm{SG}}^2}$,*

$$\int_{B(0,L)^c} e^{c\|X\|^2} dp(x) \leq 2\left(1 + \frac{c\|X\|_{\mathrm{SG}}^2}{1 - c\|X\|_{\mathrm{SG}}^2}\right) e^{-L^2 \cdot \left(\frac{1}{\|X\|_{\mathrm{SG}}^2} - c\right)}.$$

(*iii*) *For any $L > 0$,*

$$\int_{B(0,L)^c} \|x\| dp(x) \leq \left(2L + \frac{\|X\|_{\mathrm{SG}}^2}{L}\right) e^{-\frac{L^2}{\|X\|_{\mathrm{SG}}^2}}.$$

*Proof.* (*i*) As in Vershynin (2018, Prop. 2.5.2), we have

$$\mathbb{P}(\|X\| \geq s) = \mathbb{P}\left(e^{\frac{\|X\|^2}{\|X\|_{\mathrm{SG}}^2}} \geq e^{\frac{s^2}{\|X\|_{\mathrm{SG}}^2}}\right) \leq e^{-\frac{s^2}{\|X\|_{\mathrm{SG}}^2}} \mathbb{E}\left[e^{\frac{\|X\|^2}{\|X\|_{\mathrm{SG}}^2}}\right] \leq 2e^{-\frac{s^2}{\|X\|_{\mathrm{SG}}^2}},$$

where the first inequality follows from Markov inequality and the second uses the definition of norm-subgaussianity. Using (A.1), we obtain

$$\mathbb{E}\left[\|X\|^2\right] = \int_0^\infty \mathbb{P}\left(\|X\|^2 \geq k\right) dk \leq 2 \int_0^\infty e^{-\frac{k}{\|X\|_{\mathrm{SG}}^2}} dk = 2\|X\|_{\mathrm{SG}}^2,$$

which proves (A.2).

(*ii*) Let $Y = e^{c\|X\|^2} \mathbb{1}_{\{\|X\| \geq L\}}$. Then, $\int_{B(0,L)^c} e^{c\|X\|^2} p(x) dx = \mathbb{E}[Y]$. Moreover, the following chain of inequalities holds:

$$\begin{aligned}
\mathbb{E}[Y] &= \int_0^\infty \mathbb{P}(Y \geq k) dk \\
&\leq e^{cL^2} \cdot \mathbb{P}(\|X\| \geq L) + \int_{e^{cL^2}}^\infty \mathbb{P}\left(\|X\|^2 \geq \frac{\log k}{c}\right) dk \\
&\leq 2e^{cL^2} e^{-\frac{L^2}{\|X\|_{\mathrm{SG}}^2}} + 2 \int_{e^{cL^2}}^\infty k^{-\frac{1}{c\|X\|_{\mathrm{SG}}^2}} dk \\
&= 2\left(1 + \frac{c\|X\|_{\mathrm{SG}}^2}{1 - c\|X\|_{\mathrm{SG}}^2}\right) e^{-L^2\left(\frac{1}{\|X\|_{\mathrm{SG}}^2} - c\right)},
\end{aligned}$$

where the third line follows from point (*i*).

(*iii*) Similarly to the proof of the previous point, let $Y = \|X\| \mathbb{1}_{\|X\| \geq L}$. Then,

$$\int_{B(0,L)^c} \|X\| p(x) dx = \mathbb{E}[Y].$$

Moreover, the following chain of inequalities holds:

$$\mathbb{E}[Y] = \int_0^\infty \mathbb{P}(Y \geq k)\, dk$$

$$\leq L \cdot \mathbb{P}(\|X\| \geq L) + \int_L^\infty \mathbb{P}(\|X\| \geq k)\, dk$$

$$\leq 2L e^{-\frac{L^2}{\|X\|_{\mathrm{SG}}^2}} + 2 \int_L^\infty e^{-\frac{k^2}{\|X\|_{\mathrm{SG}}^2}}\, dk$$

$$\leq 2L e^{-\frac{L^2}{\|X\|_{\mathrm{SG}}^2}} + \frac{\|X\|_{\mathrm{SG}}^2}{L} e^{-\frac{L^2}{\|X\|_{\mathrm{SG}}^2}}$$

$$= \left(2L + \frac{\|X\|_{\mathrm{SG}}^2}{L}\right) e^{-\frac{L^2}{\|X\|_{\mathrm{SG}}^2}},$$

where the second inequality follows from point (*i*) and the third one follows from Lemma A.2.

$\square$

The next lemma will be useful to find an upper bound for $\epsilon_{\mathrm{MGF}}$ in terms of $b(T_1)$; the corresponding lower bound is easy and follows immediately from Jensen's inequality.

**Lemma A.4.** *Consider a random variable $F$ such that $0 \leq F \leq M$ for some $M > 0$. Then,*

$$\mathbb{E}\left[e^F\right] \leq 1 + e^M \mathbb{E}[F].$$

*Proof.* By the mean value theorem, for $x \geq 0$ we have the bound $e^x \leq 1 + x e^x$. Hence, $e^F \leq 1 + e^M F$, from which the conclusion follows by taking the expectation on both sides. $\square$

## B  Proof of Theorem 4.1

We begin with the proof of Proposition 4.3, which gives $W_2$-estimates for the reverse process $Y_t$ satisfying (4.2). Its time-marginals $q_t = \mathrm{law}(Y_t)$ satisfy the continuity equation

$$\begin{cases} q_0 = \mathrm{law}(Z_{T_2}), \\ \partial_t q_t(x) + \nabla \cdot [q_t(x)(x + \widehat{s}_\theta(t,x))] = 0. \end{cases} \tag{B.1}$$

Here and below we use the symbol $\widehat{(\cdot)}$ to denote the time-reversal of a function on $[0, T_1]$, e.g., $\hat{p}_t = p_{T_1-t}$, $\widehat{s}_\theta(t,\cdot) = s_\theta(T_1 - t, \cdot)$, $\hat{b}(t) = b(T_1 - t)$, $\widehat{L_s}(t) = L_s(T_1 - t)$.

Our goal is to obtain an upper bound for $W_2(p_\tau, q_{T_1-\tau})$, where $(\hat{p}_t)_t$ satisfies the Fokker–Planck equation

$$\begin{cases} \hat{p}_0 = p_{T_1}, \\ \partial_t \hat{p}_t(x) + \nabla \cdot [\hat{p}_t(x)(x + \nabla \log \hat{p}_t(x))] = 0. \end{cases} \tag{B.2}$$

Following Kwon et al. (2022), we apply the following well-known formula for the derivative of the Wasserstein distance between two curves of probability measures, cf. Ambrosio et al. (2008, Thm. 8.4.7, Rmk. 8.4.8), Villani (2009, Thm. 23.9).

**Theorem B.1.** *Let $(\mathcal{P}_2(\mathbb{R}^d), W_2)$ be the space of probability measures on $\mathbb{R}^d$ with finite second moment equipped with the Wasserstein distance $W_2$. Consider two weakly continuous curves $(\mu_t)_t, (\nu_t)_t$ in $\mathcal{P}_2(\mathbb{R}^d)$ that solve the continuity equations*

$$\partial_t \mu_t + \nabla \cdot (\xi_t \mu_t) = 0, \qquad \partial_t \nu_t + \nabla \cdot \left(\tilde{\xi}_t \nu_t\right) = 0.$$

*Suppose moreover that, for some $0 \leq t_1 < t_2 < \infty$, we have*

$$\int_{t_1}^{t_2} \left( \mathbb{E}_{\mu_t}\left[ \|\xi_t\|^2 \right] + \mathbb{E}_{\nu_t}\left[ \|\tilde{\xi}_t\|^2 \right] \right) dt < \infty.$$

*Then, denoting by $\pi_t$ an optimal coupling for $W_2(\mu_t, \nu_t)$, we have*

$$\frac{d}{dt} \frac{W_2^2(\mu_t, \nu_t)}{2} = \mathbb{E}_{\pi_t}\left[ (x - y) \cdot (\xi_t(x) - \tilde{\xi}_t(y)) \right],$$

*for a.e. $t \in (t_1, t_2)$.*

To apply the theorem above and deduce a differential inequality, we first need to prove the following result.

**Lemma B.2.** *For $0 < \tau < T_1 < \infty$, we have*

$$\int_0^{T_1 - \tau} \mathbb{E}_{q_t}\left[ \|x + \widehat{s}_\theta(t, x)\|^2 \right] dt < \infty, \tag{B.3}$$

$$\int_0^{T_1 - \tau} \mathbb{E}_{\hat{p}_t}\left[ \|x + \nabla \log \hat{p}_t(x)\|^2 \right] dt < \infty. \tag{B.4}$$

*Hence, the curves $(q_t)_{t \in [0, T_1 - \tau]}$ and $(\hat{p}_t)_{t \in [0, T_1 - \tau]}$ are absolutely continuous in $\left( \mathcal{P}_2(\mathbb{R}^d), W_2 \right)$.*

*Proof.* We start with (B.4). Recall first that for $t > 0$,

$$-\frac{d}{dt} D_{\mathrm{KL}}(p_t \,\|\, \gamma) = \mathcal{I}_\gamma(p_t) = \mathbb{E}_{p_t}\left[ \left\| \nabla \log \frac{dp_t}{d\gamma} \right\|^2 \right] = \mathbb{E}_{p_t}\left[ \|x + \nabla \log p_t(x)\|^2 \right],$$

where, with abuse of notation, we have identified the probability measures $p_t, \gamma$ with their densities with respect to the Lebesgue measure. Integrating this inequality between $\tau$ and $T_1$ we find

$$\int_\tau^{T_1} \mathbb{E}_{p_t}\left[ \|x + \nabla \log p_t(x)\|^2 \right] dt \leq D_{\mathrm{KL}}(p_\tau \,\|\, \gamma) < \infty,$$

where we used non-negativity of the KL-divergence and that $D_{\mathrm{KL}}(p_\tau \,\|\, \gamma) < \infty$, cf. Villani (2003, Rmk. 9.4). The conclusion follows by a change of variable in the integral.

As for (B.3), we argue as in the proof of Villani (2009, Thm. 23.9). Note first that $q_0 \in \mathcal{P}_2(\mathbb{R}^d)$, since $p_{T_1} \in \mathcal{P}_2(\mathbb{R}^d)$ and $W_2(q_0, p_{T_1}) < \infty$. Let $v_t(x)$ denote the velocity field $v_t(x) = x + \widehat{s}_\theta(t, x)$. Since $T_1$ is finite, it follows from our assumption (A1) in Section 4 that there exists a constant $C > 0$ such that $\|v_t(x)\| \leq C(1 + \|x\|)$ for all $x \in \mathbb{R}^s$ and $0 \leq t \leq T_1$. The Lipschitz assumption on $s_\theta$ in our assumption (A1) in Section 4 also implies that $v$ is Lipschitz. Therefore, there exists a unique trajectory map $T_t \colon \mathbb{R}^d \to \mathbb{R}^d$ associated to the continuity equation (B.1), i.e.,

$$\begin{cases} T_0(x) &= x, \\ \frac{d}{dt} T_t(x) &= v_t(T_t(x)). \end{cases}$$

Then, by the conservation of mass formula (Villani, 2009), we have $q_t = (T_t)_\# q$, where $\#$ denotes the pushforward of a measure by a map. Notice now that, for all $0 \leq t \leq T_1$,

$$\|T_t(x)\| = \left\| x + \int_0^t v_t(T_t)(x) dt \right\| \leq \|x\| + CT_1 + C \int_0^t \|T_t(x)\| dt.$$

Therefore, by the integral version of Gronwall's lemma applied to the continuous function $t \to \|T_t(x)\|$, we deduce that

$$\|T_t(x)\| \leq (\|x\| + CT_1) e^{CT_1}.$$

It follows that, for $0 \leq t \leq T_1$,

$$\int_{\mathbb{R}^d} \|x\|^2 q_t(dx) = \int_{\mathbb{R}^d} \|T_t(x)\|^2 q_0(dx) \leq e^{2CT_1} \int_{\mathbb{R}^d} (\|x\| + CT_1)^2 q_0(dx) =: \tilde{C} < \infty,$$

thus the second moment of $q_t$ is uniformly bounded by $\tilde{C}$ for $0 \leq t \leq T_1$. Replacing $\tilde{C}$ with $\tilde{C} + 1$, we note that also the first moment is uniformly bounded by $\tilde{C}$. Therefore, recalling that $\|x + s_\theta(t, x)\| = \|v_{T_1-t}(x)\| \leq C(1 + \|x\|)$, we obtain the following bound, for all $0 \leq t \leq T_1$,

$$\mathbb{E}_{q_t}\left[\|x + s_\theta(t, x)\|^2\right] \leq C(1 + \mathbb{E}_{q_t}\left[\|x\|^2\right]) \leq C(1 + \tilde{C}).$$

This implies the desired estimate (B.3).

Finally, the absolute continuity of the curves $(q_t)_{t \in [0, T_1 - \tau]}$ and $(\hat{p}_t)_{t \in [0, T_1 - \tau]}$ is an immediate consequence of the bounds (B.3) and (B.4) in view of Ambrosio et al. (2008, Thm. 8.3.1). $\qquad\square$

*Proof of Proposition 4.3.* Thanks to Lemma B.2, we can apply Theorem B.1. Let $\pi_t$ be an optimal coupling in $W_2$ for $\hat{p}_t$ and $q_t$, so that by definition we have $\mathbb{E}_{\pi_t}\left[\|x - y\|^2\right] = W_2^2(\hat{p}_t, q_t)$. Then, we deduce that, for a.e. $t \in [0, T_1 - \tau]$,

$$\begin{aligned}
&\frac{1}{2}\frac{d}{dt}W_2^2(\hat{p}_t, q_t) \\
&= \mathbb{E}_{\pi_t}[(x - y) \cdot (x - y)] + \mathbb{E}_{\pi_t}[(x - y) \cdot (\nabla \log \hat{p}_t(x) - \widehat{s_\theta}(t, y))] \\
&= W_2^2(\hat{p}_t, q_t) + \mathbb{E}_{\pi_t}[(x - y) \cdot (\widehat{s_\theta}(t, x) - \widehat{s_\theta}(t, y))] + \mathbb{E}_{\pi_t}[(x - y) \cdot (\nabla \log \hat{p}_t(x) - \widehat{s_\theta}(t, x))] \\
&\leq W_2^2(\hat{p}_t, q_t) + \widehat{L_s}(t)\mathbb{E}_{\pi_t}\left[\|x - y\|^2\right] + \sqrt{\mathbb{E}_{\pi_t}[\|x - y\|^2]} \cdot \sqrt{\mathbb{E}_{\pi_t}[\|\nabla \log \hat{p}_t(x) - \widehat{s_\theta}(t, x)\|^2]} \\
&= \left(1 + \widehat{L_s}(t)\right)W_2^2(\hat{p}_t, q_t) + \sqrt{\hat{b}(t)}W_2(\hat{p}_t, q_t).
\end{aligned}$$

From this we deduce the differential inequality

$$\frac{d}{dt}W_2(\hat{p}_t, q_t) \leq \left(1 + \widehat{L_s}(t)\right)W_2(\hat{p}_t, q_t) + \sqrt{\hat{b}(t)}.$$

We can solve this differential inequality by introducing the auxiliary function $I(\tau, t) := \exp\left(t - \tau + \int_\tau^t L_s(r)dr\right)$, which satisfies

$$I(\tau, r)I(r, t) = I(\tau, t), \quad I(t, t) = 1, \quad \text{and} \quad \frac{d}{dt}I(\tau, t) = (1 + L_s(t))I(\tau, t). \tag{B.5}$$

Combining the latter identity with the differential inequality above, we find

$$\frac{d}{dt}\left(I(T_1, T_1 - t)W_2(\hat{p}_t, q_t)\right) \leq I(T_1, T_1 - t)\sqrt{\hat{b}(t)},$$

for a.e. $t$. Since the curve $t \to I(\tau, t)$ is Lipschitz by (A1) in Section 4, it is also absolutely continuous. Moreover, the triangle inequality for $W_2$ yields

$$\begin{aligned}
|W_2(\hat{p}_t, q_t) - W_2(\hat{p}_s, q_s)| &\leq |W_2(\hat{p}_t, q_t) - W_2(\hat{p}_t, q_s)| + |W_2(\hat{p}_t, q_s) - W_2(\hat{p}_s, q_s)| \\
&\leq W_2(q_t, q_s) + W_2(\hat{p}_t, \hat{p}_s).
\end{aligned}$$

Using the absolute continuity of $\hat{p}_t, q_t$ in $(\mathcal{P}_2(\mathbb{R}^d), W_2)$ (cf. Lemma B.2) we deduce that $t \to W_2(\hat{p}_t, q_t)$ is absolutely continuous too on $[0, T_1 - \tau]$. Therefore, also the function

$$t \to I(T_1, T_1 - t)W_2(\hat{p}_t, q_t)$$

is absolutely continuous on $[0, T_1 - \tau]$. Hence, we can apply the second fundamental theorem of calculus for the Lebesgue integral and integrate the differential inequality between 0 and $T_1 - \tau$. Doing this gives

$$I(T_1, \tau)W_2(p_\tau, q_{T_1 - \tau}) \leq W_2(p_{T_1}, q_0) + \int_0^{T_1 - \tau} I(T_1, T_1 - t)\sqrt{\hat{b}(t)}dt.$$

Using the properties of $I$ from (B.5) we find

$$W_2(p_\tau, q_{T_1-\tau}) \leq I(\tau, T_1) W_2(p_{T_1}, q_0) + \int_0^{T_1-\tau} I(\tau, T_1 - t)\sqrt{\hat{b}(t)}dt,$$

which yields the desired expression after a change of variables $t' := T_1 - t$. $\qquad \square$

We now prove Lemma 4.4, which gives a well-known Hölder continuity bound in Wasserstein distance for the Ornstein-Uhlenbeck flow.

*Proof of Lemma 4.4.* Let $X$ be a random variable with law $p$, and let be $Z$ be a standard Gaussian random variable that is independent of $X$. Then $X_\tau := e^{-\tau}X + \sqrt{1 - e^{-2\tau}}Z$ has law $p_\tau$. Using independence, we obtain

$$W_2^2(p, p_\tau) \leq \mathbb{E}\big[|X - X_\tau|^2\big] = \mathbb{E}\Big[|(1 - e^{-\tau})X - \sqrt{1 - e^{-2\tau}}Z|^2\Big]$$
$$= (1 - e^{-\tau})^2 \mathbb{E}\big[|X|^2\big] + (1 - e^{-2\tau})\mathbb{E}\big[|Z|^2\big] \leq \tau^2 M^2 + 2\tau d,$$

which implies the result. $\qquad \square$

As discussed in Section 4.1, to establish fast convergence of the approximate Langevin dynamics in (4.1) we need a quantitative estimate for the log-Sobolev constant of $p_{T_1}$, which is provided in the next result.

**Lemma B.3.** *Let $(p_t)_{t \geq 0}$ be the law of the Ornstein–Uhlenbeck flow starting from a norm-subgaussian random vector $X$. Then, $p_t$ satisfies a log-Sobolev inequality with constant*

$$C_{\mathrm{LS}}(p_t) \geq \frac{1}{1 + 172\|X\|_{\mathrm{SG}}^2 e^{-2t}},$$

*for all $t > t_0 := \frac{1}{2}\log(1 + 4\|X\|_{\mathrm{SG}}^2)$.*

*Consequently, for any $\delta \in (0, 1)$, the log-Sobolev constant of $p_t$ satisfies $C_{\mathrm{LS}}(p_t) \geq 1 - \delta$ whenever $t \geq \max\big(t_0, \frac{1}{2}\log(172\|X\|_{\mathrm{SG}}^2/\delta)\big)$.*

The proof is based on the following recent result from Chen et al. (2021a, Thm. 2), which gives an estimate for the log-Sobolev constant of Gaussian convolutions of sub-Gaussian distributions.

**Theorem B.4.** *Let $\mu$ be a probability measure and $\sigma, C_{\mathrm{SG}} > 0$ be such that*

$$\int \int e^{\frac{\|x - x'\|^2}{\sigma^2}} \mu(dx)\mu(dx') \leq C_{\mathrm{SG}}. \tag{B.6}$$

*For all $t > \sigma^2$, the measure $\mu * \gamma_t$ satisfies a log-Sobolev inequality with the constant*

$$C_{\mathrm{LS}}(\mu * \gamma_t) \geq \left(3t\left[\frac{t}{t - \sigma^2} + C_{\mathrm{SG}}^{\frac{\sigma^2}{t - \sigma^2}}\right]\left[1 + \frac{\sigma^2}{t - \sigma^2}\log C_{\mathrm{SG}}\right]\right)^{-1}. \tag{B.7}$$

*Proof of Lemma B.3.* Note that $p_t$ is the law of $e^{-t}X + \sqrt{1 - e^{-2t}}Z$, with $X \sim p$ and $Z \sim \gamma$ independent. Consequently, $p_t = \mu_t * \gamma_{1-e^{-2t}}$, where $\mu_t$ denotes the law of $e^{-t}X$. Suppose now that $1 + 4\|X\|_{\mathrm{SG}}^2 < e^{2t}$ and define $\sigma := \sqrt{2}e^{-t}\|X\|_{\mathrm{SG}}$. Since $1 - e^{-2t} - 2\sigma^2 > 0$, we may proceed as in Chen et al. (2021a, Rmk. 3) and write

$$p_t = \mu_t * \gamma_{2\sigma^2} * \gamma_{1-e^{-2t}-2\sigma^2}. \tag{B.8}$$

We claim that $\mu_t$ satisfies the assumption of Theorem B.4 with $C_{\mathrm{SG}} = 4$ and $\sigma$ as defined above. Indeed,

$$\int_{\mathbb{R}^d} \int_{\mathbb{R}^d} e^{\frac{\|x - x'\|^2}{\sigma^2}} \mu_t(dx)\mu_t(dx') \leq \int_{\mathbb{R}^d} \int_{\mathbb{R}^d} e^{2\frac{\|x\|^2 + \|x'\|^2}{\sigma^2}} \mu_t(dx)\mu_t(dx')$$
$$= \left(\int_{\mathbb{R}^d} e^{\frac{2\|x\|^2}{\sigma^2}} \mu_t(dx)\right)^2 \leq 4, \tag{B.9}$$

where the last step uses our definition of $\sigma$. Therefore, an application of Theorem B.4 yields

$$C_{\mathrm{LS}}(\mu_t * \gamma_{2\sigma^2}) \geq \left[6\sigma^2(2+C)(1+\log C)\right]^{-1} \geq \left[86\sigma^2\right]^{-1}.$$

Using the subadditivity of $C_{\mathrm{LS}}^{-1}$ under convolution (cf. Wang & Wang (2016, Prop. 1.1)) and the estimate $C_{\mathrm{LS}}(\gamma_{1-e^{-2t}-2\sigma^2}) \geq C_{\mathrm{LS}}(\gamma) = 1$ from Lemma A.1, we obtain using (B.8),

$$C_{\mathrm{LS}}(p_t) \geq \left[\frac{1}{C_{\mathrm{LS}}(\mu_t * \gamma_{2\sigma^2})} + \frac{1}{C_{\mathrm{LS}}(\gamma_{1-e^{-2t}-2\sigma^2})}\right]^{-1} \geq \left[86\sigma^2 + 1\right]^{-1},$$

which proves the first part of the statement. The second part follows immediately. $\square$

*Proof of Lemma 4.8.* We proceed in two steps. Suppose first that $p = \delta_x$ for some $x \in \mathbb{R}^d$. Then $p_t = \gamma_{e^{-t}x,\sigma_t}$ is Gaussian with $\sigma_t := 1 - e^{-2t}$. An explicit calculation gives

$$D_{\mathrm{KL}}\left(\gamma \,\|\, \gamma_{e^{-t}x,\sigma_t}\right) = \frac{d}{2\sigma_t}\left(e^{-2t}\frac{\|x\|^2}{d} + \sigma_t \log \sigma_t - \sigma_t + 1\right). \tag{B.10}$$

In the general case where $p \in \mathbb{R}^d$ has finite second moment, we condition on the initial value using the disintegration formula

$$p_t(dy) = \int_{\mathbb{R}^d} \gamma_{e^{-t}x,\sigma_t}(dy)\, p(dx).$$

Using this formula, we employ the joint convexity of the KL-divergence and (B.10) to obtain

$$D_{\mathrm{KL}}(\gamma \,\|\, p_{T_1}) = D_{\mathrm{KL}}\left(\gamma \,\Big\|\, \int_{\mathbb{R}^d} \gamma_{e^{-t}x,\sigma_t}\, dp(x)\right) = D_{\mathrm{KL}}\left(\int_{\mathbb{R}^d} \gamma\, dp(x) \,\Big\|\, \int_{\mathbb{R}^d} \gamma_{e^{-t}x,\sigma_t}\, dp(x)\right)$$

$$\leq \int_{\mathbb{R}^d} D_{\mathrm{KL}}\left(\gamma \,\|\, \gamma_{e^{-t}x,\sigma_t}\right) dp(x) = \frac{d}{2\sigma_t}\left(\frac{M_2(p)}{d}e^{-2t} + \sigma_t \log \sigma_t - \sigma_t + 1\right),$$

where $M_2(p) := \int \|x\|^2\, dp(x)$.

For the second claim, we use the scalar inequalities $r \log r - r + 1 \leq (r-1)^2$ for $r \geq 0$. Thus, whenever $e^{-2t} \leq \frac{1}{2}$, we have

$$D_{\mathrm{KL}}(\gamma \,\|\, p_{T_1}) \leq \frac{d}{2(1-e^{-2t})}\left(\frac{M_2(p)}{d}e^{-2t} + e^{-4t}\right) \leq \left(M_2(p) + \frac{d}{2}\right)e^{-2t}.$$

This implies the desired result. $\square$

We are now ready to prove Theorem 4.1.

*Proof of Theorem 4.1.* Note first that, by the triangle inequality for $W_2$, we have

$$W_2(p, p_\theta) = W_2(p, q_{T_1-\tau}) \leq W_2(p, p_\tau) + W_2(p_\tau, q_{T_1-\tau}).$$

The first term can be estimated with Lemma 4.4, the second with Proposition 4.3. Plugging in these estimates gives

$$W_2(p, p_\theta) \leq \sqrt{3d\tau} + I_\tau(T_1)W_2(p_{T_1}, q_0) + \int_\tau^{T_1} I_\tau(t)\sqrt{b(t)}dt.$$

Therefore, to prove (4.3), it suffices to show that

$$W_2(p_{T_1}, q_0) \leq \sqrt{\frac{2}{1-\delta}\left(\delta e^{-\frac{(1-\delta)T_2}{2}} + 2\epsilon_{\mathrm{MGF}}\right)}.$$

To see this, observe that $C_{\mathrm{LS}}(p_{T_1}) \geq 1 - \delta$ by Lemma B.3: therefore, we can combine (4.12) with Theorem 4.7 to deduce that

$$W_2(p_{T_1}, q_0) \leq \sqrt{\frac{2}{1-\delta}\Big(D_{\mathrm{KL}}(\gamma \,\|\, p_{T_1})e^{-\frac{(1-\delta)T_2}{2}} + 2\epsilon_{\mathrm{MGF}}\Big)}.$$

Recalling that $\mathbb{E}\big[\|X\|^2\big] \leq 2\|X\|_{\mathrm{SG}}^2$ by $(i)$ of Lemma A.3, we can use the estimate $D_{\mathrm{KL}}(\gamma \,\|\, p_{T_1})$ from Lemma 4.8 to prove (4.3); an application of Cauchy–Schwarz inequality then gives (4.4). Finally, if we only know $T_1 \geq \frac{1}{2}\log\Big(2 + 172\frac{\|X\|_{\mathrm{SG}}^2}{\delta}\Big)$, then we can instead estimate $D_{\mathrm{KL}}(p_{T_1} \,\|\, \gamma) \leq \frac{d}{3}$, again by Lemma 4.8, which proves our claim in the discussion on the role of $T_1$ after Theorem 4.1. $\qquad\square$

## C   Proof of Theorem 4.2

When starting an Ornstein–Uhlenbeck flow from a norm-subgaussian distribution, the distribution at time $T_1$ will be norm-subgaussian too, and we can estimate its norm.

**Lemma C.1.** *Let $(X_t)_{t\geq 0}$ be an Ornstein–Uhlenbeck process* (3.6) *starting from a norm-subgaussian random vector $X$. Then, if $T_1 \geq \log\frac{\|X\|_{\mathrm{SG}}}{\sqrt{d}}$, we have*

$$\|X_{T_1}\|_{\mathrm{SG}} \leq 3\sqrt{d}. \tag{C.1}$$

*Proof.* Let $Z \sim \gamma$ be independent of $X$. Then, $X_{T_1}$ is equal in law to $e^{-T_1}X + \sqrt{1 - e^{-2T_1}}Z$. Consequently,

$$\|X_{T_1}\|_{\mathrm{SG}} = \Big\|e^{-T_1}X + \sqrt{1 - e^{-2T_1}}Z\Big\|_{\mathrm{SG}} \leq e^{-T_1}\|X\|_{\mathrm{SG}} + \|Z\|_{\mathrm{SG}} \leq 3\sqrt{d},$$

where in the last inequality we use Lemma A.1 and the choice of $T_1$. $\qquad\square$

The following lemma gives an a priori estimate for $\nabla \log p_{T_1}$ which can be used to correct predictions of $s_\theta(T_1, \cdot)$ that are far from the ground-truth.

**Lemma C.2.** *Let $(p_t)_{t\geq 0}$ be the law of the Ornstein–Uhlenbeck flow starting from a norm-subgaussian random vector $X$ with law $p$. Fix $0 < \delta < 1$ and take $T_1 \geq \log\big(\frac{16}{\delta}d(\|X\|_{\mathrm{SG}} + 1)\big)$. Then, for all $x \in \mathbb{R}^d$ and $i \in \{1, \ldots, d\}$, we have*

$$|-x_i - \partial_i \log p_{T_1}(x)| \leq \frac{\delta}{2d}(1 + \|x\|).$$

*Proof.* Let $\mu$ be the law of $e^{-T_1}X$ and set $\sigma^2 = 1 - e^{-2T_1}$. By our choice of $T_1$, we have

$$\big\|e^{-T_1}X\big\|_{\mathrm{SG}} \leq \frac{\delta}{16d} \quad \text{and} \quad e^{-2T_1} = 1 - \sigma^2 \leq \Big(\frac{\delta}{16d}\Big)^2. \tag{C.2}$$

Notice that $p_{T_1} = \mu * \gamma_{\sigma^2}$. Therefore, for all $x \in \mathbb{R}^d$, we can write as in Bardet et al. (2018)

$$p_{T_1}(x) = \int_{\mathbb{R}^d} (2\pi\sigma^2)^{-\frac{d}{2}} \exp\Big(-\frac{\|x - z\|^2}{2\sigma^2}\Big)\mu(dz) = (2\pi\sigma^2)^{-\frac{d}{2}} \exp\Big(-\Big(\frac{\|x\|^2}{2\sigma^2} + W_\sigma(x)\Big)\Big),$$

where we set

$$W_\sigma(x) = -\log \int_{\mathbb{R}^d} \exp\Big(\frac{x \cdot z}{\sigma^2} - \frac{\|z\|^2}{2\sigma^2}\Big)\mu(dz).$$

Taking the logarithm and differentiating, we find that

$$\partial_i \log p_{T_1}(x) = -\frac{1}{\sigma^2}x_i - \partial_i W_\sigma(x). \tag{C.3}$$

Observe now that

$$\left|\sigma^2 \partial_i W_\sigma(x)\right| \leq \frac{\int_{\mathbb{R}^d} |z_i| \exp\left(\frac{x\cdot z}{\sigma^2} - \frac{\|z\|^2}{2\sigma^2}\right)\mu(dz)}{\int_{\mathbb{R}^d} \exp\left(\frac{x\cdot z}{\sigma^2} - \frac{\|z\|^2}{2\sigma^2}\right)\mu(dz)} = \frac{\int_{\mathbb{R}^d} |z_i|\gamma_{x,\sigma^2}(z)\mu(dz)}{\int_{\mathbb{R}^d} \gamma_{x,\sigma^2}(z)\mu(dz)}, \tag{C.4}$$

where, with some abuse of notation, $\gamma_{x,t}$ denotes the density of a gaussian $\mathcal{N}(x, tI_d)$. We claim that

$$\int_{\mathbb{R}^d} |z_i|\gamma_{x,\sigma^2}(z)\mu(dz) \leq \frac{\delta(1+\|x\|)}{4d} \int_{\mathbb{R}^d} \gamma_{x,\sigma^2}(z)\mu(dz), \tag{C.5}$$

which we prove later. Using this bound in (C.4) we deduce that

$$|\partial_i W_\sigma(x)| \leq \frac{\delta(1+\|x\|)}{4d\sigma^2}.$$

Inserting this estimate in (C.3), it follows using (C.2) that

$$|-\partial_i \log p_{T_1}(x) - x_i| \leq \frac{1-\sigma^2}{\sigma^2}|x_i| + \frac{\delta(1+\|x\|)}{4d\sigma^2} \leq \frac{1+\|x\|}{\sigma^2}\left[\left(\frac{\delta}{16d}\right)^2 + \frac{\delta}{4d}\right] \leq \frac{\delta}{2d}(1+\|x\|),$$

where in the last inequality we used that

$$\frac{1}{\sigma^2}\left[\left(\frac{\delta}{16d}\right)^2 + \frac{\delta}{4d}\right] \leq \frac{1}{1-e^{-2T_1}}\left[\frac{1}{256} + \frac{1}{4}\right]\frac{\delta}{d} \leq \frac{1}{1-1/256}\frac{\delta}{3d} \leq \frac{\delta}{2d},$$

since $0 < \delta < 1$ and $T_1 \geq \log(16)$. This the desired estimate.

It remains to prove (C.5). To do so, we start by writing

$$\begin{aligned}
\int_{\mathbb{R}^d} \gamma_{x,\sigma^2}(z)\mu(dz) &\geq \int_{B(0,\delta)} \gamma_{x,\sigma^2}(z)\mu(dz) \\
&\geq (2\pi\sigma^2)^{-\frac{d}{2}} \exp\left(-\frac{(\|x\|+\delta)^2}{2\sigma^2}\right)\mu(B(0,\delta)) \\
&\geq (2\pi\sigma^2)^{-\frac{d}{2}} \exp\left(-\frac{(\|x\|+\delta)^2}{2\sigma^2}\right)[1 - 2\exp(-256)],
\end{aligned} \tag{C.6}$$

where in the last step we use $(i)$ of Lemma A.3 and the estimate $\|e^{-T_1}X\|_{\mathrm{SG}} \leq \frac{\delta}{16d} \leq \frac{\delta}{16}$, which holds in view of (C.2). We now split the integral in the left-hand side of (C.5) into two terms that we will estimate separately. Set $r = \frac{\delta(1+\|x\|)}{8d}$. Then,

$$\int_{B(0,r)} |z_i|\gamma_{x,\sigma^2}(z)\mu(dz) \leq r\int_{B(0,r)} \gamma_{x,\sigma^2}(z)\mu(dz) \leq \frac{\delta(1+\|x\|)}{8d}\int_{\mathbb{R}^d} \gamma_{x,\sigma^2}(z)\mu(dz).$$

Therefore, recalling (C.6), to conclude the proof of (C.5) it is enough to show that

$$\int_{B(0,r)^c} |z_i|\gamma_{x,\sigma^2}(z)\mu(dz) \leq \frac{\delta(1+\|x\|)}{8d}[1 - 2\exp(-256)](2\pi\sigma^2)^{-\frac{d}{2}}\exp\left(-\frac{(\|x\|+\delta)^2}{2\sigma^2}\right).$$

To do this, we write

$$\begin{aligned}
(2\pi\sigma^2)^{\frac{d}{2}}\int_{B(0,r)^c} |z_i|\gamma_{x,\sigma^2}(z)\mu(dz) &\leq \int_{B(0,r)^c} \|z\|\mu(dz) \\
&\leq \left(2r + \frac{\delta^2}{256d^2r}\right)\exp\left(-\frac{256d^2r^2}{\delta^2}\right) \\
&= \left(\frac{2\delta}{8d}(1+\|x\|) + \frac{\delta}{32d}\frac{1}{1+\|x\|}\right)\exp\left(-4(1+\|x\|)^2\right) \\
&\leq \frac{3\delta}{8d}(1+\|x\|)\exp\left(-4(1+\|x\|)^2\right),
\end{aligned}$$

where in the second line we have used *(iii)* of Lemma A.3. As $\sigma^2 \geq \frac{1}{2}$ by (C.2), $\delta < 1$, and $3\exp(-4) \leq [1 - 2\exp(-256)]$, we have

$$3\exp\big(-4(1 + \|x\|)^2\big) \leq [1 - 2\exp(-256)]\exp\left(-\frac{(\|x\| + \delta)^2}{2\sigma^2}\right)$$

for all $x$, which concludes the proof. $\qquad\square$

We are now ready to move to the proof of Theorem 4.2.

*Proof of Theorem 4.2.* Notice that, thanks to Lemma C.2 and to our definition of $\widetilde{s}_\theta$, for all $x \in \mathbb{R}^d$ we have

$$\|\nabla \log p_{T_1}(x) - \widetilde{s}_\theta(x)\|^2 \leq \frac{\delta^2}{d}(1 + \|x\|)^2, \tag{C.7}$$

$$\|\nabla \log p_{T_1}(x) - \widetilde{s}_\theta(x)\| \leq \|\nabla \log p_{T_1}(x) - s_\theta(T_1, x)\|. \tag{C.8}$$

As $\log(1 + \epsilon) \leq \epsilon$, it suffices to show that

$$\int_{\mathbb{R}^d} \exp\big(\beta\|\nabla \log p_{T_1}(x) - \widetilde{s}_\theta(x)\|^2\big)\,dp_{T_1}(x) \leq 1 + \epsilon.$$

Now let us fix a radius $R > 0$, whose value we specify later. We will show that, for an appropriate choice of $R > 0$,

$$\int_{B(0,R)^c} \exp\big(\beta\|\nabla \log p_{T_1}(x) - \widetilde{s}_\theta(x)\|^2\big)\,dp_{T_1}(x) \leq \frac{\epsilon}{2}, \tag{C.9}$$

and

$$\int_{B(0,R)} \exp\big(\beta\|\nabla \log p_{T_1}(x) - \widetilde{s}_\theta(x)\|^2\big)\,dp_{T_1}(x) \leq 1 + \frac{\epsilon}{2}, \tag{C.10}$$

thus concluding the proof.

First, we consider (C.9). Notice that

$$\begin{aligned}
\exp\big(\beta\|\nabla \log p_{T_1}(x) - \widetilde{s}_\theta(x)\|^2\big) &\leq \exp\left(\beta\frac{\delta^2}{d}(1 + \|x\|)^2\right) \\
&\leq \exp\left(\frac{2\beta\delta^2}{d}\right)\exp\left(\frac{2\beta\delta^2}{d}\|x\|^2\right) \\
&\leq \left(1 + \frac{4\beta\delta^2}{d}\right)\exp\left(\frac{2\beta\delta^2}{d}\|x\|^2\right) \\
&\leq 2\exp\left(\frac{1}{18d}\|x\|^2\right).
\end{aligned}$$

Here, for the first inequality we use (C.7); for the third inequality, we use that $e^s \leq 1 + 2s$ for $0 \leq s \leq 1$ and the condition on $\beta$; the condition on $\beta$ is used again for the last inequality. Therefore, we deduce that

$$\begin{aligned}
\int_{B(0,R)^c} \exp\big(\beta\|\nabla \log p_{T_1}(x) - \widetilde{s}_\theta(x)\|^2\big)\,dp_{T_1}(x) &\leq 2\int_{B(0,R)^c} \exp\left(\frac{1}{18d}\|x\|^2\right)\,dp_{T_1}(x) \\
&\leq 8e^{-\frac{R^2}{18d}},
\end{aligned}$$

where for the last inequality we use *(ii)* of Lemma A.3 and Lemma C.1. Therefore, picking

$$R = \sqrt{18d\log\frac{16}{\epsilon}}$$

readily gives (C.9). With this choice of $R$, we now consider (C.10). Let us define the random variable

$$F = \beta\|\nabla \log p_{T_1}(X_{T_1}) - \widetilde{s}_\theta(X_{T_1})\|^2\mathbb{1}_{\big\{\|X_{T_1}\|\leq R\big\}},$$

where $X_{T_1} \sim p_{T_1}$ as usual. We note that

$$\int_{B(0,R)} \exp\left(\beta\|\nabla \log p_{T_1} - \widetilde{s}_\theta\|^2\right) p_{T_1}(x)dx \leq \mathbb{E}\left[e^F\right].$$

It remains to estimate $\mathbb{E}\left[e^F\right]$, which we will do using Lemma A.4. To show that $F$ satisfies the conditions of Lemma A.4, we check its boundedness. Using (C.7) and the constraint on $\beta$ we obtain

$$0 \leq F \leq \beta\frac{\delta^2}{d}(1+R)^2 \leq \frac{1}{18d} + 36\beta\delta^2 \log\frac{16}{\epsilon} =: M.$$

Furthermore, using (C.8) and (4.7) we can estimate $\mathbb{E}[F]$ by

$$\mathbb{E}[F] \leq \beta\mathbb{E}_{p_{T_1}}\left[\|\nabla \log p_{T_1} - s_\theta(T_1, \cdot)\|^2\right] \leq \frac{1}{34}\epsilon^{1+36\beta\delta^2}.$$

Notice also that

$$e^M = e^{\frac{1}{18d}}16^{36\beta\delta^2}\epsilon^{-36\beta\delta^2} \leq 16e^{\frac{1}{18}}\epsilon^{-36\beta\delta^2},$$

where we used once more the constraint on $\beta$. We can now apply Lemma A.4 to deduce that

$$\mathbb{E}\left[e^F\right] - 1 \leq e^M\mathbb{E}[F] \leq \frac{16e^{\frac{1}{18}}}{34}\epsilon \leq \frac{\epsilon}{2},$$

which concludes the proof. $\qquad\square$

## D   Proof of Theorem 5.1

The first step of the argument in the proof of Theorem 5.1 consists in giving an upper bound for $\text{TV}(p, p_\theta)$ which allows to control separately the errors originating from *(i)* taking $Z_{N_2}$ as an approximate sample from $p_{T_1}$, and *(ii)* approximating the reverse process with a discretized scheme and with an $L^2$ accurate score. To do so, we follow the strategy of Chen et al. (2023b; 2022a). Let us denote by $S$ the Markov kernel which associates to a probability measure $\mu$ the law of the random variable $U_{T_1}$, where $(U_t)_t$ satisfies the true backward SDE initialised at $\mu$, *i.e.*,

$$\begin{cases} U_0 \sim \mu, \\ dU_t = U_t dt + 2\nabla \log p_{T_1-t}(U_t)dt + \sqrt{2}dB_t. \end{cases}$$

In particular, we have $p = p_{T_1}S$. Similarly, we denote by $\hat{S}$ the Markov kernel which corresponds to following the approximate reverse process in (5.2) initialised at $\mu$. In particular, we have $p_\theta = \sigma_{N_2}\hat{S}$ (recall that $\sigma_k = \text{law}(Z_k)$). The following chain of inequalities holds:

$$\begin{aligned} \text{TV}(p, p_\theta) &= \text{TV}(p_{T_1}S, \sigma_{N_2}\hat{S}) \\ &\leq \text{TV}(p_{T_1}S, p_{T_1}\hat{S}) + \text{TV}(p_{T_1}\hat{S}, \sigma_{N_2}\hat{S}) \\ &\lesssim \sqrt{D_{\text{KL}}\left(p_{T_1}S \,\middle\|\, p_{T_1}\hat{S}\right)} + \text{TV}(p_{T_1}, \sigma_{N_2}) \\ &\lesssim \sqrt{D_{\text{KL}}\left(p_{T_1}S \,\middle\|\, p_{T_1}\hat{S}\right)} + \sqrt{D_{\text{KL}}(\sigma_{N_2} \,\|\, p_{T_1})}. \end{aligned} \tag{D.1}$$

In the above, we have used the triangle inequality for the total variation distance, Pinsker's inequality and the data-processing inequality. This achieves the desired decomposition, so that we can study the two processes (5.1), (5.2) separately.

**Convergence of inexact Langevin algorithm.**

To control the error term $D_{\mathrm{KL}}(\sigma_{N_2} \| p_{T_1})$, we need to study convergence of the process (5.1). This is done by adapting the results of Yingxi Yang & Wibisono (2023) to the case of a decaying step size. In particular, we prove the following

**Proposition D.1.** *Let Assumption (A2) hold, pick $0 < \delta < \frac{1}{2}$, $T_1 \geq \frac{1}{2} \log\left(2 + 172 \frac{\|X\|_{\mathrm{SG}}^2}{\delta} + \frac{d}{2\delta}\right)$ and suppose in addition that $\nabla \log p_{T_1}$ is $L_1$-Lipschitz and that $s_\theta(T_1, \cdot)$ is $L_2$-Lipschitz, with $L_1, L_2 \geq 1$. Then, for the inexact Langevin algorithm (5.1) with step sizes $h_k = \frac{1}{24 L_1 L_2 + \frac{k+1}{16}}$, we have that*

$$D_{KL}(\sigma_{N_2} \| p_{T_1}) \lesssim \left(\frac{L_1 L_2}{N_2 + 1}\right)^2 + \frac{d L_2^2}{N_2 + 1} + \tilde{\epsilon}_{\mathrm{MGF}}, \tag{D.2}$$

*where $\tilde{\epsilon}_{\mathrm{MGF}}$ is defined in (5.3).*

The proof is postponed to the end of this section. Using Proposition D.1 gives the desired upper bound for the second error term in (D.1), when using a decaying step size. For the analogous result with a constant step size, see (Yingxi Yang & Wibisono, 2023, Thm. 2).

**Analysis of the reverse process.**

It remains to give an upper bound for the error due to the discretization and approximation of the score in the reverse process, corresponding to the first term in the right hand side of (D.1). This has been analysed in a number of recent works, under different assumptions. We recall in particular the following results, proved in Chen et al. (2022a) building on the Girsanov framework developed in Chen et al. (2023b).

**Lemma D.2.** *Suppose that $p$ has finite second moment and that $\nabla \log p_t$ is $L_1$-Lipschitz for $t \in [0, T_1]$ with $T_1, L_1 \geq 1$. Then, choosing a constant step size $\tilde{h}_k = \frac{T_1}{N_1} \leq 1$ gives*

$$D_{KL}\left(p \,\middle\|\, p_{T_1} \hat{S}\right) = D_{KL}\left(p_{T_1} S \,\middle\|\, p_{T_1} \hat{S}\right) \lesssim \widehat{J_{SM}} + \frac{d L^2 T_1^2}{N_1}.$$

Inserting this bound in (D.1) concludes the proof of Theorem 5.1.

**Proof of Proposition D.1**

The proof of Proposition D.1 is based on the results of Yingxi Yang & Wibisono (2023), and in particular on Lemma 6 therein, which upper bounds the relative entropy after one step of the inexact Langevin algorithm and which we recall below.

**Lemma D.3** (Lemma 6 of Yingxi Yang & Wibisono (2023))**.** *Let $\mu, \nu_0$ be probability measures with full support that admit densities with respect to the Lebesgue measure. Suppose that $\mu$ satisfies a log-Sobolev inequality and let $0 < \kappa \leq C_{\mathrm{LS}}(\mu)$. In addition, suppose that $s_\theta$ is an approximation of $\nabla \log \mu$, that $s_\theta$ is $L_2$-Lipschitz and that $\nabla \log \mu$ is $L_1$-Lipschitz with $L_1, L_2 \geq 1$. Set $Z_0 \sim \nu_0$ and*

$$Z_1 = Z_0 + h s_\theta(Z_0) + \sqrt{2h} B$$

*where $B \sim \mathcal{N}(0, I_d)$ is independent of $Z_0$ and $0 < h < \min\left\{\frac{\kappa}{12 L_1 L_2}, \frac{1}{2\kappa}\right\}$. Then, letting $\nu_1 = \mathrm{law}(Z_1)$, we have*

$$D_{KL}(\nu_1 \| \mu) \leq e^{-\frac{1}{4}\kappa h} D_{KL}(\nu_0 \| \mu) + 144 L_2^2 L_1 d h^3 + 24 d L_2^2 h^2 + h \frac{\kappa}{2} \log \mathbb{E}_\mu\left[\frac{9}{\kappa} \|s_\theta - \nabla \log \mu\|^2\right].$$

With this lemma at hand, we can prove Proposition D.1.

*Proof of Proposition D.1.* By the choice of $T_1$, we have that $C_{\mathrm{LS}}(p_{T_1}) \geq 1 - \delta \geq \frac{1}{2}$. In particular, the step sizes $(h_k)_{k=0}^{N_2-1}$ defined by

$$h_k = \frac{1}{24L_1L_2 + \frac{k+1}{16}}$$

satisfy the constraint in Lemma D.3 with $\mu = p_{T_1}$ and $s_\theta = s_\theta(T_1, \cdot)$. We can therefore apply the lemma to deduce that, after each step of the inexact Langevin algorithm,

$$D_{\mathrm{KL}}(\sigma_{k+1} \,\|\, p_{T_1}) \leq e^{-\frac{1}{8}h_k} D_{\mathrm{KL}}(\sigma_k \,\|\, \mu) + 30dL_2^2 h_k^2 + h_k \tilde{\epsilon}_{\mathrm{MGF}},$$

for $0 \leq k \leq N_2 - 1$. By iterating the above result we find that

$$D_{\mathrm{KL}}(\sigma_{N_2} \,\|\, p_{T_1}) \leq e^{-\frac{1}{8}\sum_{i=0}^{N_2-1} h_i} + \sum_{i=0}^{N_2-1} \left\{ \left(30dL_2^2 h_i^2 + h_i \tilde{\epsilon}_{\mathrm{MGF}}\right) \cdot e^{-\frac{1}{8}\sum_{j=i+1}^{N_2-1} h_j} \right\}, \tag{D.3}$$

where we have also used that $D_{\mathrm{KL}}(\sigma_0 \,\|\, p_{T_1}) \leq \delta \leq 1$ by Lemma 4.8.

Now, notice that, for $0 \leq j \leq k$, we have the bound

$$\sum_{i=j}^{k} h_i \geq \int_j^{k+1} \frac{1}{24L_1L_2 + \frac{x+1}{16}} dx = 16 \log \frac{24L_1L_2 + \frac{k+2}{16}}{24L_1L_2 + \frac{j+1}{16}} \tag{D.4}$$

and

$$1 > \frac{h_{k+1}}{h_k} \geq \frac{h_1}{h_0} = \frac{24L_1L_2 + \frac{1}{16}}{24L_1L_2 + \frac{1}{8}} \geq \frac{1}{2}.$$

Using this in (D.3), we find that

$$\begin{aligned}
D_{\mathrm{KL}}(\sigma_{N_2} \,\|\, p_{T_1}) &\leq \left( \frac{24L_1L_2 + \frac{1}{16}}{24L_1L_2 + \frac{N_2+1}{16}} \right)^2 + \sum_{i=0}^{N_2-1} \left\{ \left(30dL_2^2 h_i^2 + h_i \tilde{\epsilon}_{\mathrm{MGF}}\right) \cdot \left( \frac{24L_1L_2 + \frac{i+2}{16}}{24L_1L_2 + \frac{N_2+1}{16}} \right)^2 \right\} \\
&\leq \left( \frac{2}{1 + \frac{N_2}{384L_1L_2}} \right)^2 + 30dL_2^2 \sum_{i=0}^{N_2-1} h_i^2 \frac{h_{N_2}^2}{h_{i+1}^2} + \tilde{\epsilon}_{\mathrm{MGF}} \sum_{i=0}^{N_2-1} h_i \cdot \frac{h_{N_2}}{h_{i+1}} \\
&\leq \left( \frac{2}{1 + \frac{N_2}{384L_1L_2}} \right)^2 + 120dL_2^2 N_2 \cdot \left( \frac{1}{24L_1L_2 + \frac{N_2}{16}} \right)^2 + 2\tilde{\epsilon}_{\mathrm{MGF}} \cdot \frac{N_2}{24L_1L_2 + \frac{N_2}{16}} \\
&\leq \left( \frac{2}{1 + \frac{N_2}{384L_1L_2}} \right)^2 + 30720 \frac{dL_2^2}{N_2 + 1} + 32\tilde{\epsilon}_{\mathrm{MGF}} \\
&\lesssim \left[ \left( \frac{L_1L_2}{N_2 + 1} \right)^2 + \frac{dL_2^2}{N_2 + 1} + \tilde{\epsilon}_{\mathrm{MGF}} \right].
\end{aligned} \tag{D.5}$$

This concludes the proof of the proposition. $\qquad \square$

