# OpenReview forum: "Improved Convergence of Score-Based Diffusion Models via Prediction-Correction"
_TMLR — Accepted by TMLR_

### Review · Reviewer_y515 · 2024-02-05

**Summary Of Contributions:**

This manuscript presents a novel approach to enhancing the convergence of score-based diffusion models by integrating a predictor-corrector scheme that operates within a fixed finite time horizon, $T_1$. Instead of starting the reverse process at a Gaussian, this scheme proposes to start the reverse process at a sample from the final distribution of the forward process, obtained via Langevin dynamics with estimated score. A significant technical contribution is the derivation of error bounds that vanish as both the discretization step size and the score matching error approach zero, even for the specified fixed time horizon $T_1$. This development is grounded in recent theoretical advancements in diffusion models and maintains polynomial-time guarantees with minimal assumptions, primarily requiring the score function to be accurately learned in $L^2$.

**Audience:**

Yes

**Broader Impact Concerns:**

None.

**Claims And Evidence:**

Yes

**Requested Changes:**

Motivation: Prior results already only required $T_1$ to be logarithmic in problem parameters, which seems quite mild. In light of this, it would be helpful to further discuss the importance of taking $T_1$ fixed, especially since the present work only seems to shave off a $\log(1/\epsilon)$ factor in the choice of $T_1$ anyway.

Training vs. Generation: The paper makes a compelling point that reducing $T_1$ leads to smaller score matching error, which is important for training the diffusion model. But another important aspect is the cost of generation itself, for which the main important quantity is the number of evaluations of the neural network. In this regard, although the proposed algorithm shrinks $T_1$, the corrector steps require additional function evaluations, and it is unclear whether this leads to an overall speed-up for generation. Therefore, it is worth remarking that the main implications of the paper are for training, not for generation.

I find Footnote 2 to be a bit inaccurate: there are general tools, such as weighted CKP inequalities or Wasserstein control via weighted TV, which suggest that control in information divergences is typically stronger than Wasserstein control.

Thm. 4.1: please comment on the apparent exponential dependence on the one-sided Lipschitz constant.

Lem. A.6: I believe this can be improved by using the Taylor expansion of the exponential, $\exp(x) \le 1 + x\exp(x)$ for any $x \ge 0$. It is an improvement because it leads to the score matching error only needing to be of size $\epsilon^{1+O(\delta^2)}$, not $\epsilon^{2+O(\delta^2)}$ (and this seems quite important since the accuracy of learning the score is the least well-understood part of the whole theory).

**Strengths And Weaknesses:**

Overall, this is a solid contribution and merits acceptance. I only have very minor comments.

**Strengths:**

Novel Approach: The idea to start the reverse process at the output of inexact Langevin is an elegant, intriguing, and novel contribution to the field of generative modeling. The methodology suggests improvements which could potentially have practical implications.

Technical Rigor: The analysis is thorough and builds on recent theoretical tools introduced for diffusion models, offering strong guarantees under practical assumptions.

**Weaknesses:**

Motivation: The manuscript would benefit from slightly more motivation for the importance of keeping $T_1$ as a fixed time horizon, see below.

---

> ### Author Response · Authors · 2024-03-07
> **Part 1**
>
> We thank the reviewer for the positive feedback and the careful read of the paper.
>
>
> **Motivation: Prior results already only required $T_1$ to be logarithmic in problem parameters, which seems quite mild. In light of this, it would be helpful to further discuss the importance of taking $T_1$ fixed, especially since the present work only seems to shave off a $\log(1/\epsilon)$ factor in the choice of $T_1$ anyway.**
>
>
>
> **Training vs. Generation: The paper makes a compelling point that reducing $T_1$ leads to smaller score matching error, which is important for training the diffusion model. But another important aspect is the cost of generation itself, for which the main important quantity is the number of evaluations of the neural network. In this regard, although the proposed algorithm shrinks $T_1$, the corrector steps require additional function evaluations, and it is unclear whether this leads to an overall speed-up for generation. Therefore, it is worth remarking that the main implications of the paper are for training, not for generation.**
>
>
>
>
> We thank the reviewer for the suggestion. In the revised version, we clarify that one of the main benefits of having a fixed $T_1$ lies indeed in the reduced computational cost for the training of the neural network. We believe that this also sends an interesting message, which extends the conclusion of [1]: in the absence of strong functional inequalities for the target distribution $p$ (e.g. a log-Sobolev inequality), efficient sampling can still be achieved by accurately learning the score of the perturbed distribution $p_t$ on a finite times interval $[0,T_1]$, whose length depends only on the properties of $p$ (i.e., its subgaussian norm and the dimension of the space on which it lives).
>
> In terms of cost of generations, while we agree that we do not prove a substantial speed-up, we would like to remark that the discussion in Section 5 shows that the total number of evaluations of the neural network is not bigger than in other existing results for the classical reverse SDE (e.g. Theorem 2 in [1]), and in fact we actually obtain a mild improvement of $\log^2(1/\epsilon)$ in Theorem 5.1/Remark 5.3. A more important advantage of having a fixed $T_1$ in our results however is that the error bounds are stable, i.e. they do not explode as $T_2\to \infty$, as opposed to what happens when $T_1\to \infty$ in, e.g, Theorem 2 in [1].
>
>
> **I find Footnote 2 to be a bit inaccurate: there are general tools, such as weighted CKP inequalities or Wasserstein control via weighted TV, which suggest that control in information divergences is typically stronger than Wasserstein control.**
>
> What we mean is that, without additional assumptions, it is not possible to translate the available convergence guarantees in KL divergence to convergence guarantees in Wasserstein distance. Consider the following example, where $\nu = \delta_0$ plays the role of the data distribution (and is clearly subgaussian), while $\mu = (1-\epsilon)\delta_0 + \epsilon \delta_{\frac{1}{\epsilon}}$ plays the role of the output distribution.
> Then $D_{KL}(\nu||\mu) = \log \frac{1}{1-\epsilon}$ converges to $0$ as $\epsilon \to 0$, while $W_2(\nu,\mu) = \sqrt{\frac{1}{\epsilon}}$ converges to $\infty$. Therefore, the convergence guarantees in reverse KL-divergence (i.e. the bounds on $D_{KL}(p||p_\theta)$)  available in the literature cannot be translated to convergence guarantees in $W_2$ without additional assumptions (such as an artificial projection step of the algorithm, which has also the downside of worsening the dependence of the bounds on the problem parameters).

---

> ### Author Response · Authors · 2024-03-07
> **Part 2**
>
> **Thm. 4.1: please comment on the apparent exponential dependence on the one-sided Lipschitz constant.**
>
>
>
> We argue that the appearance of the one-sided Lipschitz constant in the exponential is not problematic under realistic assumptions, for the following reasons. Denote by $L_p(t)$ and  $L_s(t)$ the one-sided Lipschitz constant of $\nabla \log p_t$ and $s_\theta(t,\cdot)$ respectively.
> Let us focus first on $L_p$, corresponding to the situation where the score function is known exactly: while $L_p(t)$ can behave badly for very small times $t$, after a small time we expect it to improve thanks to the regularizing properties of the Ornstein–Uhlenbeck flow. For example, this is true when $p$ has bounded support, in which case the one-sided Lipschitz constant can be effectively bounded (and in particular becomes negative after a time which is logarithmic in the size of the support), cf. Lemma 20(3) in [1] for example. Therefore, the integral of the one-sided Lipschitz constant of $\nabla \log p_t$ is typically not too big.
> Under the more realistic assumptions that $s_\theta(t,\cdot) \approx \nabla \log p_t$ only with $L^2$-accuracy, it can be too restrictive to assume that $L_s(t) \approx L_p(t)$ for all times (or that $L_s(t) \to -1$ as $t\to \infty$, as was assumed in [2]): however, it is reasonable that, as the approximation of the score improves, then on average $L_s$ does not behave too differently from $L_p$, in which case we expect the integral $\int_\tau^{T_1} L_s(t) dt$ to be of the same order.
>
>
>
>
>
>
>
> **Lem. A.6: I believe this can be improved by using the Taylor expansion of the exponential, $\exp(x) \le 1+x \exp(x)$ for any $x \ge 0$. It is an improvement because it leads to the score matching error only needing to be of size $\epsilon^{1+O(\delta^2)}$, not  $\epsilon^{2+O(\delta^2)}$, (and this seems quite important since the accuracy of learning the score is the least well-understood part of the whole theory).**
>
> We thank the reviewer for this useful remark. We agree that this indeed improves the bound, as it translates to a milder requirement on the accuracy of the score estimate at time $T_1$. We have updated the Lemma (and Theorem 4.2) in the revised version and acknowledged the reviewer for the suggestion.
>
>
> **References:**
>
> [1] Sitan Chen, Sinho Chewi, Jerry Li, Yuanzhi Li, Adil Salim, and Anru Zhang.
> *Sampling is as easy as learning the score: theory for diffusion models with minimal data assumptions.*
> In International Conference on Learning Representations (ICLR), 2023b.
>
> [2] Dohyun Kwon, Ying Fan, and Kangwook Lee. *Score-based generative modeling secretly minimizes the
> Wasserstein distance.* In Advances in Neural Information Processing Systems (NeurIPS), 2022.

---

### Review · Reviewer_XHhW · 2024-02-15

**Summary Of Contributions:**

This work introduces new convergence guarantees for a variant of the popular predictor-corrector approach in score-based generative models. As a theoretical study, its main technical contribution lies in offering convergence guarantees that do not require an infinite time (i.e., $T_1<\infty$) for the forward process. Moreover, it establishes these guarantees under only minimal and realistic assumptions on the data distribution and score estimation error.

**Audience:**

Yes

**Claims And Evidence:**

Yes

**Requested Changes:**

Overall I find the paper well-written and very clear. I have some questions/comments below.

I think the choice of T2 (and N2 in the discretized case) is crucial for the convergence results, and also useful for practical implementations.  It is better to have a more concrete comment regarding the choice or guidance on T2 in Section 4. Related to this, it would also be useful to comment on the tradeoff between the optimal choice of N1 and N2 from eq (5.4).

From Theorem 4.1, it can be seen that we do need $T_2\to \infty$ in order to ensure vanishing sampling error. Since large $T_2$ also results in an increased computational complexity of the generation process, I wonder how the authors would comment and compare the complexity of the original diffusion models and the current method via Prediction-Correction.

From eq (4.4), will it be possible to comment on the optimum choice of early stopping time $\tau$? (as there is a clear tradeoff between the first term (increasing function of $\tau$) and the second term (decreasing function of $\tau$).

**Strengths And Weaknesses:**

Strength: The presented convergence result is new, and the balance between the finite time steps for the two phases -- (i) the inexact Langevin to generate initial values, and (ii) the reverse process -- could be of importance and interest in the machine learning community. This could also provide theoretical guidance for practical implementation.

Weakness: I didn't see any notable flaws. This paper could be enhanced by incorporating additional discussion and interpretations for the theoretical findings, as commented below.

---

> ### Author Response · Authors · 2024-03-07
>
> **Requested changes:**
>
> **I think the choice of T2 (and N2 in the discretized case) is crucial for the convergence results, and also useful for practical implementations. It is better to have a more concrete comment regarding the choice or guidance on T2 in Section 4. Related to this, it would also be useful to comment on the tradeoff between the optimal choice of N1 and N2 from eq (5.4).**
>
> **From Theorem 4.1, it can be seen that we do need $T_2 \to \infty$ in order to ensure vanishing sampling error. Since large $T_2$ also results in an increased computational complexity of the generation process, I wonder how the authors would comment and compare the complexity of the original diffusion models and the current method via Prediction-Correction.**
>
>
>
> Thank you for these comments. Indeed, letting $T_2\to \infty$ is needed for convergence (because this is needed in general for Langevin dynamics, even when targeting log-concave distributions and with perfect knowledge of the score). To better understand the computational complexity of the method compared to the standard reverse SDE without corrector step, we believe that it is more meaningful to look at discretizations of the algorithm and the discussion in Section 5. By adapting existing analysis, we show that we can obtain convergence guarantees for the two-stage method comparable to the existing one for the standard reverse diffusion: in fact, this also shows a mild logarithmic improvement with respect to the desired sampling accuracy (see Section 5.1). This shows that in general the predictor-corrector method performs at least as well as following the reverse SDE without correction. In practice, predictor-corrector methods are often used, and numerical evidence suggests that the speed-up could be more significant (see [3, Section 4.2]).
> However, we remark that the main advantages of the two-stage predictor-corrector method that we studied lie not in the speed of sampling, but rather in the reduced computational complexity for learning the score and in the stability of the convergence results. In fact, by keeping $T_1$ fixed, we can see that the proposed error estimates do not diverge even if we let $T_2 \to \infty$.
>
>
>
> **From eq (4.4), will it be possible to comment on the optimum choice of early stopping time $\tau$? (as there is a clear tradeoff between the first term (increasing function of $\tau$) and the second term (decreasing function of $\tau$).**
>
>
> We thank the reviewer for this question. For general data distributions, without any further assumption, an early stopping time (i.e. $\tau>0$) is useful. This is because $\nabla \log p_t$ can be ill-behaved for small times $t$, which causes instability towards the end of the reverse process (3.3). In fact, early stopping is frequently used in practice [3, Sec. C].
> The consequence of early stopping is that instead of sampling from $p$ we sample from a slightly perturbed distribution $p_\tau$, and this gives a first source of error, which we estimate by $\sqrt{3d\tau}$ in Wasserstein distance, and which appears in (4.3), (4.4).
> In practice, this suggests that one should choose first the desired sampling accuracy $\epsilon$ and then $\tau$ accordingly, so that the difference between $p$ and $p_\tau$ is irrelevant (if we measure errors in $W_2$-distance, (4.3), (4,4) suggests to choose $\tau$ so that $\sqrt{3d\tau}\lesssim \epsilon$).
> In some situations, when the distribution $p$ is regular enough, it might be that early stopping is not needed, so we keep the possibility to choose $\tau =0$ in our bounds.
> For convergence guarantees in reverse KL-divergence for a discrete scheme, for example, it was recently proved in [4] that $\tau = 0$ can be chosen provided that $p$ has finite Fisher-information with respect to the Gaussian distribution, relaxing therefore the assumptions in previous work. Notice however that this assumption is not satisfied by some distributions, for example under the manifold hypothesis, i.e. when $p$ is supported on a strictly lower-dimensional manifold.
>
>
>
>
> **References:**
>
> [3] Yang Song, Jascha Sohl-Dickstein, Diederik P Kingma, Abhishek Kumar, Stefano Ermon, and Ben Poole.
> *Score-Based Generative Modeling through Stochastic Differential Equations.*
> In: International Conference on Learning Representations (ICLR). 2021.
>
> [4] Giovanni Conforti, Alain Durmus, and Marta Gentiloni Silveri. *Score diffusion models without early stopping: finite Fisher information is all you need.*
> arXiv preprint arXiv:2308.12240, 2023.

---

### Review · Reviewer_Qg7R · 2024-03-02

**Summary Of Contributions:**

The paper introduces a novel approach to enhancing score-based generative models (SGMs). In traditional score-based generative modeling, the process involves simulating a diffusion process that gradually adds noise to the data until a simple, known distribution is reached, and then learning to reverse this process to generate data from noise. However, this approach faces challenges because the forward diffusion process is run for a finite time.

The document proposes a prediction-correction scheme to address these challenges. This method consists of two main steps: prediction, where an inexact Langevin dynamics is used to estimate the final distribution after a finite forward process; and correction, where this estimate is refined to more accurately represent the target distribution. This approach is designed to improve the efficiency and accuracy of the reverse generation process, enabling better convergence properties compared to existing methods.

The theoretical foundation of the proposed method is analyzed in the paper. It establishes improved convergence guarantees with explicit dependence on the input dimension, the early stopping time, the subgaussian norm of the target distribution, the approximation of the score, under minimal assumptions. These guarantees are significant because they allow for a fixed finite time \(T_1\) for the forward process, diverging from the traditional requirement of \(T_1 \rightarrow \infty\).

Moreover, the paper compares the proposed prediction-correction approach with previous works in the field, highlighting its advantages in terms of theoretical and practical performance. It discusses how the method provides a better trade-off between the precision of score approximation and computational costs, which is crucial for the practical application of SGMs in generating complex data samples.

Hence, the paper addresses key limitations in the convergence of traditional score-based methods, offering a more efficient and theoretically sound approach to generating high-quality samples from complex distributions. This work contributes to the theoretical understanding of SGMs but also opens up new possibilities for their application in various domains.

**Audience:**

Yes

**Claims And Evidence:**

No

**Requested Changes:**

* make a table summarizing all the notations
* I would have liked the authors to have gone a little further in their work. Let's start with simple examples, such as mixtures of sub-Gaussian laws in high dimensions - starting simply from a mixture of Gaussian, with exact score). It would be interesting to see what the theory tells us about the choice of hyperparameters $T_1$, $T_2$, etc... And it would then be interesting to understand whether the theory at least partially reflects the "right" choices of these hyperparameters.
* There are many completely trivial results at the beginning of the appendix on sub-Gaussian variables. It would be simpler to gather on one page the properties of the sub-Gaussian variables you'll have to use
* Fix the proof of Proposition D.1 and Lemma D.3

**Strengths And Weaknesses:**

Strengths:
1. Introduces a novel prediction-correction scheme improving convergence in score-based generative models.
2. The paper Offers theoretical guarantees for accuracy  with minimal assumptions on the data. The theoretical guarantees depend explicity on the constants of the problem: input dimension,  subgaussian norm of the target distribution, etc.
3. Addresses computational efficiency, making it more feasible for practical applications.
4. Includes comparative analysis with previous methods, highlighting its advancements and  contributes to the theoretical understanding of the dynamics in generative modeling.

Weaknesses:
1. Theoretical analysis might not fully account for all practical scenarios and data distributions. Some assumptions are but some assumptions are rather ad-hoc and not sufficiently discussed. Is it reasonable to assume, for example, that the data are "norm-sub Gaussian"? and if that's the case [after all, all variables are bounded] what could be the value of this norm for imagenet, for example?
2. I'm not sure the paper adds anything to our existing understanding of the problem. The algorithms presented are well known, so only the analysis can be considered original. The analysis itself repeats many elements already known, and so the contribution is quite incremental [it "complements" a set of known results rather than making a decisive contribution to the subject].
3. I haven't been able to study all the proofs in detail, but it seems to me that the paper contains errors - missing assumptions. I have some doubts on the statement and the proof of Proposition D.1 which is based on a Lemma 6 of Yang and Wisibono - Some of the assumptions of Lemma 6 are not mentioned - in particular, one of the assumption in Lemma 6 is
"Assumption 5 ( $L^{\infty}$ error assumption). The error of score estimator $s(x)$ has finite $L^{\infty}$ norm at every $x$, i.e.
$$
M_{\infty}=\sup _{x \in \mathbb{R}^n}\|\nabla \log \nu(x)-s(x)\|<\infty .
$$
"
and $M_\infty$ appears in the bound of Lemma 6. In the Lemma D.3 of the present paper [which is simply presented as a restatement of Lemma 6], this assumption has disappeared and, on the top of that,  the statement of Lemma D.3 differs from the statement of Lemma 6 [the supremum norm has disappeared]

---

> ### Author Response · Authors · 2024-03-07
> **Part 1**
>
> We thank the reviewer for the comments. We address below the concerns.
>
> **Theoretical analysis might not fully account for all practical scenarios and data distributions. Some assumptions are but some assumptions are rather ad-hoc and not sufficiently discussed. Is it reasonable to assume, for example, that the data are "norm-sub Gaussian"? and if that's the case [after all, all variables are bounded] what could be the value of this norm for imagenet, for example?**
>
> If a random variable $X$ has bounded support (say contained in the euclidean ball $B(0,R)$ of radius $R>0$), then its subgaussian norm $||X||_{SG}$ is upper bounded by $R/\sqrt{2}$. For image generations, pixels are typically rescaled in $[0,1]$, hence the subgaussian norm is upper bounded by $\sqrt{d}$, where $d$ is the dimension.
> Besides bounded support, many more distributions are norm-subgaussian: for example, mixtures of subgaussian distributions are also subgaussian, hence this assumption allows for complex multi-modal distributions; data produced via a Generative Adversarial Network (GAN) is subgaussian as well [7]. Moreover, it is also much weaker than requiring that the distribution satisfies a log-Sobolev inequality.
> Therefore, we believe that this assumption is realistic and covers most practical cases. In addition, notice that the dependency of $T_1$ on the subgaussian norm is only logarithmic.
>
>
>
>
>
> **I'm not sure the paper adds anything to our existing understanding of the problem. The algorithms presented are well known, so only the analysis can be considered original. The analysis itself repeats many elements already known, and so the contribution is quite incremental [it "complements" a set of known results rather than making a decisive contribution to the subject].**
>
>
> We believe our submission presents several new ideas. While there are now many theoretical papers studying the standard reverse diffusion, for predictor corrector schemes we are only aware of the concurrent work [6], which studies a different variant. Under mild assumptions comparable to ours, to the best of our knowledge our work is the only one to give theoretical convergence guarantees that require only a fixed perturbation time $T_1$. While part of the analysis is based on the existing literature, our results required also innovative ideas (e.g. in the choice of the specific variant of the predictor-corrector scheme based on the new observation that $p_{T_1}$ is appropriately regularized by the OU flow, or in the truncation procedure of Theorem 4.2) and significant technical work (e.g. in the new estimates that we use to prove Theorem 4.2, see Lemma C.2).
>
>
> **I haven't been able to study all the proofs in detail, but it seems to me that the paper contains errors - missing assumptions. I have some doubts on the statement and the proof of Proposition D.1 which is based on a Lemma 6 of Yang and Wisibono - Some of the assumptions of Lemma 6 are not mentioned - in particular, one of the assumption in Lemma 6 is "Assumption 5 ($L^\infty$ error assumption). The error of score estimator $s(x)$ has finite $L^\infty$ norm at every $x$, i.e. $ M_{\infty}=\sup_{x \in \mathbb{R}^n}|\nabla \log \nu(x)-s(x)|<\infty . $ " and $M_\infty$ appears in the bound of Lemma 6. In the Lemma D.3 of the present paper [which is simply presented as a restatement of Lemma 6], this assumption has disappeared and, on the top of that, the statement of Lemma D.3 differs from the statement of Lemma 6 [the supremum norm has disappeared].**
>
>
>
> We thank the referee for this comment. We emphasize that, to the best of our knowledge, all proofs are correct and there are no missing assumptions. The confusion is due to the fact that our bibliography lists the version of [5] that has been accepted at the NeurIPS 2022 Workshop on Score-Based Methods, while we should have listed the arxiv version of that paper, in which some theorems have different numbers: in particular, Lemma 6 in the arxiv version corresponds to Lemma 3 in the published version. As the constants are slightly different in both versions, we prefer to cite the arxiv version. We corrected the bibliography in the revised version of our manuscript.

---

> > ### Comment · Reviewer_y515 · 2024-03-07
> > **Predictor-corrector**
> >
> > I'd like to interject with a quick comment that the predictor-corrector scheme was also studied in https://arxiv.org/pdf/2206.06227.pdf.

---

> > > ### Author Response · Authors · 2024-03-08
> > >
> > > Thank you for pointing this related work. We already cite it in the current version, but we will add further clarifications in the related work section. We would like to highlight that this paper belongs to the line of work making strong assumptions on the data distribution. In particular, it assumes that the data distribution satisfies a log-Sobolev inequality, which is much more restrictive than assuming norm-subgaussianity. The log-Sobolev constant also enters crucially in the derived error bounds, which depend polynomially on it, and not just logarithmically. In addition, the predictor-corrector scheme considered is different from ours.

---

> ### Author Response · Authors · 2024-03-07
> **Part 2**
>
> **REQUIRED CHANGES:**
>
>
> **Make a table summarizing all the notations**
>
>
> We thank the reviewer for the suggestion. For the reader’s convenience, in the revision we provide a table collecting the important notations at the beginning of the appendix.
>
>
> **I would have liked the authors to have gone a little further in their work. Let's start with simple examples, such as mixtures or sub-gaussian laws in high dimensions - starting simply from a mixture of Gaussian, with exact score. It would be interesting to see what the theory tells us about the choice of hyperparameters $T_1$, $T_2$, etc… And it would then be interesting to understand whether the theory at least partially reflects the "right" choices of these hyperparameters.**
>
>
> We thank the reviewer for the suggestion. While these suggestions are interesting, we believe that a detailed study of this topic is best left for future work. In fact, even in the case of the mixture of gaussian laws, there are some non-trivial obstacles to this task, e.g. in providing accurate estimates of the one-sided Lipschitz constant (which is related to the problem of studying the Hessian of $\log p_t$ along the OU flow), or in understanding what are exactly the “right choices” of hyperparameters to be compared with our theoretical results (this would require more extended numerical experiments, which go outside the scope of the present submission).
>
>
>
> **There are many completely trivial results at the beginning of the appendix on subgaussian variables. It would be simpler to gather on one page the properties of the subgaussian variables you’ll have to use.**
>
> Thank you for the suggestion. While these results are simple, we included them in the paper for the convenience of the reader. We re-organised Appendix A by collecting the simplest results on subgaussian random variables in a single result (Lemma A.3).
>
>
> **Fix the proof of proposition D.1 and Lemma D.3.**
>
> As explained above, we have corrected the typo on the reference. Namely, Lemma D.3 in our submission corresponds to Lemma 6 of [5].
>
> **References:**
>
> [5] Kaylee Yingxi Yang and Andre Wibisono,
> *Convergence of the Inexact Langevin Algorithm and Score-based Generative Models in KL Divergence*
> arXiv preprint arXiv:2211.01512
>
> [6] Sitan Chen, Sinho Chewi, Holden Lee, Yuanzhi Li, Jianfeng Lu, and Adil Salim.
> *The probability flow ode is provably fast.*
> In: Thirty-seventh Conference on Neural Information Processing Systems, 2023
>
> [7] Mohamed El Amine Seddik, Cosme Louart, Mohamed Tamaazousti, and Romain Couillet.
> *Random matrix theory proves that deep learning representations of GAN-data behave as gaussian mixtures.*
> In International Conference on Machine Learning (ICML), 2020.

---

### Author Response · Authors · 2024-03-07
**Updated version of manuscript**

We thank the reviewers for their useful comments, which we have addressed below. We have also uploaded a revised version based on the reviewers’ suggestions, which we believe has improved the quality of the paper.
In particular:
1) We have changed Lemma A.4 (previously Lemma A.6),  as suggested by Reviewer y515, and correspondingly improved Theorem 4.2 by requiring a weaker bound in (4.7).
2) We have remarked also in the Introduction that smaller values of $T_1$ are beneficial for the computational cost of the training procedure.
3) Following Reviewer Qg7R’s suggestion, we have reorganized Appendix A, by collecting together the simple lemmas about norm-subgaussian distributions (now Lemma A.3) and by providing a table with relevant notation for the reader’s convenience.
4) We have updated the reference for Lemma D.3.

---

### Decision · Action_Editor_Xxhv · 2024-05-18

**Recommendation:** Accept as is

**Comment:**

This work studies a variant of the predictor-corrector method. The key idea is to sample from the final time step distribution via (approx) Langevin dynamics and use this sample to start a standard reverse process instead of drawing it from the stationary distribution, i.e., Gaussian. The main result provides convergence guarantees in Wasserstein distance under realistic assumptions on score estimation.

Overall, this paper offers a novel approach to diffusion models, improving practical performance with strong theoretical guarantees. The methodology of starting the reverse process at the output of inexact Langevin dynamics is innovative. The theoretical analysis is comprehensive, providing strong guarantees under mild/practical assumptions. The reviewers have highlighted the thoroughness of the analysis and its practical implications. Given these strengths, I recommend acceptance. This work is a valuable addition to the literature on diffusion models.

**Audience:**

This paper targets researchers and practitioners in the field of diffusion models. It is also relevant for those interested in the theoretical aspects of machine learning algorithms.

**Claims And Evidence:**

The authors provide a detailed theoretical analysis that includes convergence guarantees. These results are grounded in recent advancements in the field, and the assumptions made are reasonable and practical.